# Baicalein Inhibits Benzo[a]pyrene-Induced Toxic Response by Downregulating Src Phosphorylation and by Upregulating NRF2-HMOX1 System

**DOI:** 10.3390/antiox9060507

**Published:** 2020-06-09

**Authors:** Yuka Tanaka, Takamichi Ito, Gaku Tsuji, Masutaka Furue

**Affiliations:** 1Department of Dermatology, Graduate School of Medical Sciences, Kyushu University, Fukuoka 812-8582, Japan; yukat53@med.kyushu-u.ac.jp (Y.T.); takamiti@dermatol.med.kyushu-u.ac.jp (T.I.); gakku@dermatol.med.kyushu-u.ac.jp (G.T.); 2Research and Clinical Center for Yusho and Dioxin, Kyushu University Hospital, Fukuoka 812-8582, Japan; 3Division of Skin Surface Sensing, Department of Dermatology, Faculty of Medical Sciences, Kyushu University, Fukuoka 812-8582, Japan

**Keywords:** baicalein, benzo[a]pyrene, Src, aryl hydrocarbon receptor, nuclear factor-erythroid 2-related factor-2, reactive oxygen species, keratinocyte, Wogon, Oren-gedoku-to

## Abstract

Benzo[a]pyrene (BaP), a major environmental pollutant, activates aryl hydrocarbon receptor (AHR), induces its cytoplasmic-to-nuclear translocation and upregulates the production of cytochrome P450 1A1 (CYP1A1), a xenobiotic metabolizing enzyme which metabolize BaP. The BaP-AHR-CYP1A1 axis generates reactive oxygen species (ROS) and induces proinflammatory cytokines. Although the anti-inflammatory phytochemical baicalein (BAI) is known to inhibit the BaP-AHR-mediated CYP1A1 expression, its subcellular signaling remains elusive. In this study, normal human epidermal keratinocytes and HaCaT keratinocytes were treated with BAI, BaP, or BAI + BaP, and assessed for the CYP1A1 expression, antioxidative pathways, ROS generation, and proinflammatory cytokine expressions. BAI and BAI-containing herbal medicine Wogon and Oren-gedoku-to could inhibit the BaP-induced CYP1A1 expression. In addition, BAI activated antioxidative system nuclear factor-erythroid 2-related factor-2 (NRF2) and heme oxygenase 1 (HMOX1), leading the reduction of BaP-induced ROS production. The BaP-induced IL1A and IL1B was also downregulated by BAI. BAI inhibited the phosphorylation of Src, a component of AHR cytoplasmic complex, which eventually interfered with the cytoplasmic-to-nuclear translocation of AHR. These results indicate that BAI and BAI-containing herbal drugs may be useful for inhibiting the toxic effects of BaP via dual AHR-CYP1A1-inhibiting and NRF2-HMOX1-activating activities.

## 1. Introduction

Environmental pollution is a long-lasting problem for human health. Environmental pollutants such as polycyclic aromatic hydrocarbons (PAHs) contaminate the soil, food and air particles, including ambient particulate matter of up to 2.5 μm diameter (PM2.5) [1,2]. Among various PAHs, benzo[a]pyrene (BaP) accounts for 27%–67% of the toxicity of airborne particles [3]. BaP is emitted to the environment mostly from inefficient combustion of coal or from other industrial plants, and also from exhaust gas of vehicles such as automobile and airplane [4]. Once drawn into the body, BaP binds to a chemical sensor aryl hydrocarbon receptor (AHR) and is metabolized, and the metabolites trigger the following reactions, which eventually damage the cells [5,6].

AHR is a ligand-activated transcription factor and is abundantly expressed in skin cells including epidermal keratinocytes to sense environmental and endogenous chemicals [7,8,9,10]. In the absence of ligands, AHR localizes in the cytoplasm as a component of protein complex which consists of dimer of heat shock protein 90 (HSP90), hepatitis B virus X-associated protein 2 (XAP-2, also known as AIP or Ara9), p23 and c-Src protein kinase [10,11,12,13,14,15,16]. After ligand binding, XAP-2 is released from the complex and triggers conformational changes in AHR that lead to AHR nuclear translocation by exposing its nuclear localizing signal [17]. In the nucleus, AHR dimerizes with AHR-nuclear translocator (ARNT), binds DNA response elements called xenobiotic response elements (XREs) and upregulates the transcription of phase I and II xenobiotic metabolizing enzymes, such as phase I: Cytochrome P450 1A1 (CYP1A1), CYP1A2, and CYP1B1 [8,9,10,18,19] and phase II: NAD(P)H quinone reductase-1 (NQO1), UDP-glycosyltransferases (UGT1A1 and UGT1A6), and glutathione S-transferase (GST) [6,20,21,22,23]. When it binds to AHR, BaP is oxidized by CYPs into epoxides and phenolic intermediates, which are further processed by phase II enzymes. The resulting 3,6-quinone undergoes redox cycles with generation of reactive oxygen species (ROS) and leads to oxidative stress [6,24,25]. The oxidative stress increases secretion of ATP, which in turn stimulates the production of proinflammatory cytokines such as IL-1α, IL-1β, and IL-6 [7,26,27,28,29,30,31].

On the other hand, accumulating evidence suggests that certain AHR agonists such as coal tar, soybean tar, cynaropicrin and *Opuntia ficus-indica* extract are capable of inhibiting BaP-AHR-mediated oxidative stress by activating the antioxidative master transcription factor nuclear factor-erythroid 2-related factor-2 (NRF2) and subsequently inducing antioxidative enzymes such as heme oxygenase 1 (HMOX1) [32,33,34,35,36,37]. Under physiological conditions, NRF2 is localized in cytoplasm by its actin-bound inhibitor protein Kelch-like ECH-associated protein (Keap1). Oxidative stress dissociates the interaction between Keap1 and NRF2, and NRF2 commences to translocate into the nucleus where it heterodimerizes with musculoaponeurotic fibrosarcoma oncogene homolog and interacts with antioxidant response elements (ARE). Then, gene expression of the downstream target genes such as HMOX1 and NQO1 is induced and protects cells from oxidative damages [38]. HMOX1 catalyzes heme to release free iron to form biliverdin which further metabolizes to carbon monoxide (CO) and bilirubin, and exerts antioxidative effect [38]. Recent clinical trials have demonstrated that one of these antioxidative AHR agonists, tapinarof, has therapeutic potential for treating psoriasis and atopic dermatitis [39,40,41,42,43,44].

Baicalein (BAI) is one of the major active compounds found in the root of *Scutellaria baicalensis* [45,46,47]. It is also known to be a plant-derived AHR agonist with the structure of flavonoid and also the aglycone of the flavone glycoside baicalin (Figure 1). BAI is now attracting interest in the pharmaceutical, cosmetic and food industries due to its biological functions, such as anti-inflammatory, antioxidation, antibacterial and antitumor properties [48,49,50,51]. BAI is contained in the traditional crude drug Wogon (WO), which is made from the dried root of *Scutellaria baicalensis*. WO contains active compounds bailcalin and wogonin in addition to BAI, and is further mixed with other crude drugs to compose several herbal medicines, such as Oren-gedoku-to (OG, Huanglian-Jie-Du-Tang in Chinese). OG is used for alleviating the symptoms of various diseases including cerebrovascular disease, gastritis, liver dysfunction, dermatitis, inflammation, and so on [52,53,54,55].

Despite BAI being an AHR agonist and upregulating CYP1A1 expression in human HepG2 liver cancer cells [56], it paradoxically reduces the AHR-mediated CYP1A1 expression induced by 7,12-dimethylbenz[a]anthracene (DMBA) [57]. However, its subcellular mechanisms are not fully understood. In this study, it is shown that BAI ameliorates BaP-induced AHR-CYP1A1 axis by inhibiting Src phosphorylation and subsequent AHR nuclear translocation, and that BAI also inhibits the BaP-induced ROS production and proinflammatory cytokine production via induction of NRF2-HMOX1 antioxidative pathway.

## 2. Materials and Methods

### 2.1. Reagents

Dimethyl sulfoxide (DMSO; Sigma-Aldrich, St. Louis, MO, USA) was used as a solvent and was also added to culture medium as a vehicle control. BAI, BaP (both purchased from Sigma-Aldrich), dasatinib (Abcam, Cambridge, UK) and H_2_DCFDA (Thermo Fisher Scientific, Waltham, MA, USA) were all dissolved in DMSO as stock solution and further diluted in medium or buffers. The crude drug WO and the herbal medicine OG (both from Tsumura Co., Tokyo, Japan) were dissolved in distilled water at a concentration of 10 mg/mL and heated at 50 °C for 60 min. After centrifugation, supernatants were collected and added to culture medium (final concentration: 100 μg/mL).

### 2.2. Cell Culture

Normal human epidermal keratinocytes (NHEKs; Lonza, Basel, Switzerland) were maintained in KBM^TM^ Gold^TM^ Basal Medium (Lonza) supplemented with KGM^TM^ Gold^TM^ SingleQuots^TM^ supplements (Lonza). Culture media were refreshed every 2 days, and cells were passaged at sub-confluence. NHEKs were used for experiments at fewer than four passages. An immortalized human keratinocyte cell line, HaCaT (Cell Lines Service, Eppelheim, Germany), was maintained in DMEM supplemented with 10% fetal bovine serum (FBS; Sigma-Aldrich) and penicillin/streptomycin (Thermo Fisher Scientific). Culture media were refreshed every 2–3 days and cells were passaged at sub-confluence. Cells were kept in a water-jacketed CO_2_ incubator at 37 °C with 5% CO_2_.

### 2.3. Cell Proliferation Assay

NHEKs and HaCaT cells were seeded in a 96-well plate at a density of 5,000 cells/well and incubated for 24 h. They were then treated with DMSO (0.1%) or BAI (5, 10, 25, 50 or 100 μM) for 24 h. The viability of cells was determined using a Cell Counting Kit-8 (CCK-8, Dojindo Molecular Technologies, Inc., Kumamoto, Japan), in accordance with the manufacturer’s protocol. After incubation with chemicals, cells were treated with CCK-8 solution for 2–4 h which contains WST-8 coloring reagent. In live cells, WST-8 is reduced to WST-8 formazan (orange color with maximum absorption around 450 nm) and the amount of WST-8 formazan is proportional to the live cell number. Absorbance of the reaction product at 450 nm was measured using an iMark microplate reader (Bio-Rad Laboratories Inc., Hercules, CA, USA) and the results are shown as fold change compared with the DMSO-treated control.

### 2.4. Treatment of Cells for RNA and Protein Extraction

Cells were seeded at a density of 3 × 10^5^ cells/well in 6-well plates and incubated for 48 h. They were then washed twice with Dulbecco’s phosphate-buffered saline (DPBS; Fujifilm Wako Pure Chemicals Corporation, Osaka, Japan) and treated with medium containing DMSO (0.1%), BaP (1 μM), BAI (10 μM for NHEKs and 25 μM for HaCaT cells) or a combination of BaP and BAI. Cells were collected at 6 h for RNA extraction or at 12–24 h for protein extraction. Proteins of the cells were extracted with lysis buffer (25 mM HEPES, 10 mM Na_4_P_2_O_7_∙10H_2_O, 100 mM sodium fluoride, 5 mM EDTA, 2 mM Na_3_VO_4_, 1% Triton X-100) supplemented with protease inhibitor cocktail (Sigma-Aldrich).

For the experiments with herbal drugs, HaCaT cells were seeded in a 6- or 12-well plate at a density of 3 × 10^5^ or 1.5 × 10^5^ cells/well and incubated for 48 h. Cells were then washed twice with DPBS and treated with fresh medium containing DMSO (0.1%), BAI (25 μM), BaP (1 μM), WO (100 μg/mL), OG (100 μg/mL) or combinations of BaP and BAI, BaP and WO, or BaP and OG. After 6 h (for RNA extraction) and 12–24 h (for protein extraction), cells were washed twice with DPBS and collected as mentioned above.

### 2.5. RNA Extraction and Quantitative Reverse-Transcription Polymerase Chain Reaction (qRT-PCR)

RNA was extracted from cells using RNeasy Mini Kit (Qiagen, Hilden, Germany) and converted to cDNA using PrimeScript RT Reagent Kit (Takara Bio Inc., Kusatsu, Japan). PCR was performed on a CFX Connect^TM^ Real-Time System (Bio-Rad Laboratories Inc.) using TB Green Premix Ex Taq II (Takara Bio Inc.) and specific primers. The PCR amplification schedule started with 95 °C for 30 s, followed by 40 cycles of 95 °C for 5 s and 60 °C for 20 s. Expression of each target gene was normalized with the cycle threshold of β-actin (*ACTB*) and the mRNA expression relative to the DMSO-treated control sample was calculated using the comparative Ct method. The primers (Invitrogen, Carlsbad, CA, USA) used were as follows: *ACTB* (forward 5′-ATTGCCGACAGGATGCAGA-3′, reverse 5′-GAGTACTTGCGCTCAGGAGGA-3′); *AHR* (forward 5′-CAAATCCTTCCAAGCGGCATA-3′, reverse 5′-CGCTGAGCCTAAGAACTGAAAG-3′); *CYP1A1* (forward 5′-TAGACACTGATCTGGCTGCAG-3′, reverse 5′-GGGAAGGCTCCATCAGCATC-3′); *IL1A* (forward 5′-AGATGCCTGAGATACCCAAAACC-3′, reverse, 5′-CCAAGCACACCCAGTAGTCT-3′); *IL1B* (forward 5′-ATGATGGCTTATTACAGTGGCAA-3′, reverse 5′-GTCGGAGATTCGTAGCTGGA-3′); *NQO1* (forward 5′-GAAGAGCACTGATCGTACTGGC-3′, reverse 5′-GGATACTGAAAGTTCGCAGGG-3′); Glutathione Peroxidase 2 (*GPX2*) (forward 5′-GGTAGATTTCAATACGTTCCGGG-3′, reverse 5′-TAGCAGTTCTCCTGATGTCCAAA-3′).

### 2.6. siRNA Transfection

Negative control siRNA or AHR siRNA (s1200) (both purchased from Invitrogen) was diluted with RNase-free water (Thermo Fisher Scientific), further mixed with Opti-MEM^TM^ I Reduced Serum Medium (Thermo Fisher Scientific) and Lipofectamine^TM^ RNAiMAX Transfection Reagent (Thermo Fisher Scientific), and incubated for 20 min at room temperature. HaCaT cells were seeded in 12-well plates and mixed with siRNA-Lipofectamine complex (final siRNA concentration: 10 nM). After 48 h of incubation, cells were treated with DMSO (0.1%), BaP (1 μM) or BAI (25 μM) for 6 h and collected for RNA extraction. The knockdown efficiency of AHR at 48 h after transfection was determined by qRT-PCR and Western blotting.

### 2.7. ROS Measurement by Flow Cytometry

NHEKs or HaCaT cells were seeded in a 6- or 12-well plate at a density of 3 × 10^5^ or 1.5 × 10^5^ cells/well and incubated for 48 h. Cells were then washed twice with DPBS and treated with fresh medium containing DMSO (0.1%), BAI (10 μM for NHEKs and 25 μM for HaCaT cells), BaP (1 μM) or a combination of BaP and BAI. Twelve hours later, cells were washed twice with Hank’s balanced salt solution (HBSS; Fujifilm Wako Pure Chemicals Corporation, Osaka, Japan) and stained with 25 μM H_2_DCFDA (Thermo Fisher Scientific) in HBSS for 30 min at 37 °C in the dark. After washing with DPBS, cells were harvested and used for flow cytometry. Mean fluorescent intensity (MFI) of DCF dye was analyzed using FlowJo software (Tree Star, Inc., San Carlos, CA, USA). Propidium iodide (Thermo Fisher Scientific) was used to eliminate dead cells from the analysis.

### 2.8. Western Blotting

The extracted protein lysates were mixed with Sample Buffer Solution with 2-ME (2×) (Nacalai Tesque, Kyoto, Japan), heated at 96 °C for 5 min and used for SDS-PAGE on Bolt^TM^ 4%–12% Bis-Tris Plus Gels (Invitrogen). Then, the proteins were transferred to PVDF membranes (Merck Millipore, Burlington MA, USA), blocked with 2% bovine serum albumin (BSA; Sigma-Aldrich) in 0.1% Tris-Buffered Saline with Tween (0.1% TBS-T) and probed with primary specific antibodies diluted in Can Get Signal^®^ Immunoreaction Enhancer Solution 1 (Toyobo Co., Ltd., Osaka, Japan) at 4 °C overnight. The primary antibodies used were as follows: Rabbit anti-human Src antibody (#2123, 1:2000), rabbit anti-human phosphor-Src (Tyr416) antibody (#2101, 1:1000), rabbit anti-human β-actin antibody (#4970, 1:1000) (all purchased from Cell Signaling Technology, Danvers, MA, USA), rabbit anti-HO-1/HMOX1 (10701-AP, 1:1000, Proteintech, Rosemont, IL, USA), rabbit anti-CYP1A1 antibody (ab3568, 1:500, Abcam), rabbit anti-IL-1 alpha antibody (ab9614, 1:1000, Abcam), rabbit IL-1β antibody (#12703, 1:1000, Cell Signaling Technology), rabbit anti-human AHR (sc-5579, 1:1000, Santa Cruz Biotechnology, Inc., Dallas, TX, USA), and rabbit anti-human NRF2 (sc-13032, 1:1000, Santa Cruz Biotechnology, Inc.). After three washes with 0.1% TBS-T, membranes were further treated with horseradish peroxidase (HRP)-conjugated secondary antibody (HRP-linked goat anti-rabbit IgG (#7074, 1:10,000, Cell Signaling Technologies)) diluted in Can Get Signal^®^ Immunoreaction Enhancer Solution 2 (Toyobo Co., Ltd.) at room temperature for 60 min. Then, the membranes were washed three times with 0.1% TBS-T. Immunological bands were visualized with SuperSignal^TM^ West Pico Chemiluminescence Substrate (Thermo Fisher Scientific) and imaged with the ChemiDoc^TM^ XRS Plus System (Bio-Rad Laboratories Inc.), and the signals of blots were measured with Image Lab Software (Bio-Rad Laboratories Inc.). The values of whole and cytoplasmic protein (CYP1A1, NRF2, HMOX1, IL1A, IL1B, and AHR) were normalized with the value of β-ACTIN and the values of nuclear protein (Lamin B1, NRF2, and AHR) were normalized with the value of Lamin B1.

### 2.9. Extraction of Cytoplasmic and Nuclear Protein

For the analysis of AHR/NRF2 nuclear translocation by Western blotting, cytoplasmic and nuclear proteins were separately extracted using NE-PER^TM^ Nuclear and Cytoplasmic Extraction Reagents (Thermo Fisher Scientific) following its instructions. Briefly, HaCaT cells were seeded in a 6-well plate at a density of 3 × 10^5^/well and incubated for 48 h. Cells were then treated with DMSO (0.1%) or BAI (25 μM) for 6 h (for the analysis of NRF2 nuclear translocation), or were treated with DMSO (0.1%), BaP (1 μM), BaP and BAI (25 μM) or BaP and dasatinib (100 nM) for 24 h (for the analysis of AHR nuclear translocation). After incubation, cells were washed twice with DPBS, harvested using cell scraper, and centrifuged to remove excess DPBS. The cell pellets were suspended in CER I reagent and incubated for 10 min on ice. Then, CER II reagent was added and the lysates were centrifuged at 16,000× *g* at 4 °C for 10 min. The supernatants (cytoplasmic fraction) were transferred to new tubes and immediately stored at −80 °C until use. NER reagent was then added to the remaining pellet and the lysates were incubated on ice for 40 min. The lysates were mixed by vortex at every 10 min. After centrifugation at 16,000× g at 4 °C for 5 min, the supernatants (nuclear fraction) were transferred to new tubes and immediately stored at −80 °C until use.

### 2.10. Immunocytochemistry

HaCaT cells were seeded on an eight-well μ-Slide (ibidi GmbH, Martinsried, Germany) at a density of 3 × 10^4^ cells/mL (equal to 1000 cells/well) and incubated for 48 h. After two washes with DPBS, cells were treated with DMSO (0.1%) or BAI (25 μM) for 6 h (for the analysis of NRF2 nuclear translocation), or were treated with DMSO (0.1%), BaP (1 μM), BaP and dasatinib (100 nM) or BaP and BAI (25 μM) for 24 h (for the analysis of AHR nuclear translocation). After incubation, cells were fixed with cold acetone, air-dried and treated with 5% BSA to avoid non-specific antibody binding. After three washes with DPBS, fixed cells were incubated with primary antibodies at 4 °C overnight. The primary antibodies used were as follows: Rabbit anti-human AHR (sc-5579, 1:100, Santa Cruz Biotechnology, Inc.), rabbit anti-human NRF2 (sc-13032, 1:100, Santa Cruz Biotechnology, Inc.), and mouse anti-human E-cadherin (610181, 1:200, BD Biosciences, San Jose, CA, USA). Cells were then washed three times with DPBS and further treated with Alexa Fluor^®^ 546-conjugated goat anti-rabbit IgG (H + L) (A11010, 1:400, Thermo Fisher Scientific), and Alexa Fluor^®^ 488-conjugated goat anti-mouse IgG (H + L) (A11001, 1:400, Thermo Fisher Scientific) secondary antibodies in dark. After three washes with DPBS, cells were covered with Vectashield mounting medium with DAPI (Vector Laboratories, Burlingame, CA, USA) and observed with an EVOS^®^ FL fluorescent microscope (Thermo Fisher Scientific).

### 2.11. Statistical Analysis

Each experiment was performed in triplicate wells and repeated three times. Data are presented as mean ± standard deviation (SD). Statistical analyses were performed with GraphPad Prism7 software (GraphPad Software, San Diego, CA, USA). The significance of differences between two groups was determined by Student’s unpaired two-tailed *t*-test and that of three or more groups was determined by one-way ANOVA followed with multiple comparisons. Shapiro-Wilk test was used to test normality of the samples. A *p* value of less than 0.05 was considered to reflect statistical significance.

## 3. Results

### 3.1. BAI Reduces BaP-Induced CYP1A1 Expression in Keratinocytes

To investigate the effects of BAI on the *CYP1A1* expression in keratinocytes, cells were treated with BAI, BaP or both. The concentration of BaP was determined based on the previous report using the same cell type and methods of analyses, and the concentration is sufficient to activate AHR [58]. BAI did not affect cell proliferation up to 10 μM in NHEKs, and up to 25 μM in HaCaT cells, an immortalized human keratinocyte cell line [59] (Appendix A). Thus, BAI at 10 μM for NHEKs and 25 μM for HaCaT cells were used in the following experiments. As shown in Figure 2a,b, BaP significantly induced *CYP1A1* mRNA expression (10.43 ± 0.062-fold increase in NHEKs and 30.11 ± 9.04-fold increase in HaCaT cells) compared with that of the DMSO-treated control, whereas BaP-induced *CYP1A1* upregulation was significantly suppressed in the presence of BAI. Although BAI was shown to induce CYP1A1 expression in HepG2 liver cancer cells [56], it did not exhibit CYP1A1 upregulation in NHEKs (Figure 2a) but it induced only minimally in HaCaT cells, 10 times lower than that of BaP (Figure 2b). As well as mRNA, CYP1A1 protein expression was significantly induced by BaP and it was suppressed in the presence of BAI in both NHEKs and HaCaT cells (Figure 2a,b, Appendix A). Thus, BAI can suppress BaP-induced CYP1A1 expression in normal as well as immortalized keratinocytes.

In addition, since BAI is a constituent of the crude drug WO and the WO-containing herbal drug OG, it was further tested whether these phytodrugs could inhibit BaP-induced CYP1A1 expression as did BAI. WO and OG were prepared in the boiled water (at 50 °C for 60 min) and added to the culture medium with BaP. As shown in Figure 2c, both WO and OG significantly reduced the BaP-induced CYP1A1 expression at mRNA and protein levels (Figure 2c and Appendix A). WO caused stronger suppression of CYP1A1 than that of OG and its effect was comparable to that of BAI at mRNA and protein levels. These results imply that BAI and BAI-containing drugs may protect keratinocytes from the hazardous effects of BaP.

### 3.2. Baicalein Activates NRF2-HMOX1 Antioxidative Pathway

To know whether BAI is capable of activating antioxidative system, we next examined the activation, namely cytoplasmic-to-nuclear translocation, of NRF2. Immunocyochemical analysis showed that NRF2 was located in the cytoplasm in the DMSO-treated control keratinocytes, and BAI did induce its nuclear translocation (Figure 3a, arrows). NRF2 nuclear translocation was confirmed with Western blotting by separately extracting cytoplasmic and nuclear protein. As shown in Figure 3b, NRF2 protein was significantly increased in the nuclear fraction of BAI-treated cells compared to that of DMSO-treated cells (1.55 ± 0.144-fold increase, Figure 3b and Appendix A).

HMOX1 is a major downstream molecule of the antioxidative transcription factor NRF2 and activation of NRF2-HMOX1 pathway is crucial to protect keratinocytes from the oxidative damages [60,61]. To confirm HMOX1 upregulation in response to NRF2 nuclear translocation, HMOX1 expression was evaluated in keratinocytes. As shown in Figure 4, treatment with BAI significantly induced the *HMOX1* mRNA expression in NHEKs (2.06 ± 0.210-fold increase) and in HaCaT cells (1.69 ± 0.0590-fold increase). BaP did not affect the *HMOX1* expression and BAI was able to induce *HMOX1* even in the presence of BaP. The expression pattern of HMOX1 protein was consistent with that of mRNA, HMOX1 was significantly induced in BAI-treated NHEKs (1.78 ± 0.131-fold increase) and HaCaT cells (1.50 ± 0.0996-fold increase) compared to DMSO-treated control (Figure 4a,b, middle and lower panels, Appendix A). BaP did not change HMOX1 expression and BAI induced HMOX1 even in the presence of BaP (Figure 4a,b, middle and lower panels, Appendix A). Similar to BAI, BAI-containing herbal drug WO and OG also induced mRNA and protein expressions of HMOX1 (Appendix A). In addition to HMOX1, expressions of other antioxidative enzymes NQO1 and GPX2 downstream of NRF2 transcription factor were assessed. As shown in Appendix A, BAI significantly induced *NQO1* and *GPX2* in NHEKs or HaCaT keratinocytes and expression patterns of *NQO1* and *GPX2* were similar to that of *HMOX1*. WO significantly induced *NQO1* and *GPX2* and OG significantly induced *NQO1*, but not *GPX2* (Appendix A). These results indicated that BAI and BAI-containing herbal drugs were activator of NRF2 antioxidative system.

### 3.3. Baicalein Inhibits BaP-Induced ROS Production in Keratinocytes

The production of ROS is one of the major hazardous effects of BaP. Therefore, we next examined whether BAI was capable of inhibiting BaP-induced ROS production. Keratinocytes were next treated with BaP, BAI or BaP + BAI and ROS production was assessed using flow cytometry. As shown in Figure 5, BaP significantly increased the MFI (i.e., production of ROS) in NHEKs (1.42 ± 0.0214-fold higher value) and in HaCaT cells (3.40 ± 0.30-fold higher value) than that of the DMSO-treated control. However, BaP-induced ROS generation was significantly inhibited in the presence of BAI. These results indicated that BAI-mediated antioxidative signal is functional in inhibiting the BaP-induced ROS production in keratinocytes.

### 3.4. BaP-Induced Proinflammatory Cytokine Expression Is Attenuated by Baicalein

Since BaP is also known to induce proinflammatory cytokines through AHR-CYP1A1 activation and subsequent ROS production, we next examined whether BAI could prevent the BaP-induced proinflammatory cytokine expression. The expression of *IL1A* mRNA was significantly induced by BaP (1.51 ± 0.047-fold increase in NHEKs and 2.46 ± 0.13-fold increase in HaCaT cells) compared with that in the vehicle-treated control and it was significantly reduced in the presence of BAI (Figure 6a). Similar phenomenon was observed regarding *IL1B* mRNA. BaP significantly induced *IL1B* in NHEKs (1.91 ± 0.11-fold increase) and in HaCaT cells (3.45 ± 0.42-fold increase) compared with that in the control, and BAI significantly reduced the BaP-induced *IL1B* expression (Figure 6b). In accordance with mRNA expression, IL1A and IL1B proteins were significantly induced by BaP and BAI significantly suppressed the BaP-induced IL1A and IL1B protein production (Figure 6c–f, Appendix A). Single treatment of BAI significantly downregulated the mRNA/protein expressions of IL1A and IL1B in NHKEs (Figure 6). On the other hand, IL1A expression was slightly but significantly upregulated by BAI in HaCaT cells, although the level was lower than that of BaP. IL1B was not altered by BAI in HaCaT cells (Figure 6). We also assessed the AHR dependence of *IL1A* and *IL1B* induction by BaP through the knockdown of AHR using siRNA in HaCaT cells (Appendix A). Knockdown efficiency was confirmed at *AHR* mRNA (knockdown efficiency: 87.56 ± 1.69% decrease compared with control siRNA) and AHR protein levels (knockdown efficiency: 58.77 ± 15.40% decrease compared with control siRNA) (Appendix A). Once AHR was knocked down, the induction of *IL1A* and *IL1B* by BaP was inhibited, confirming the AHR-dependent induction of these cytokines (Appendix A).

### 3.5. BaP-Induced Src Phosphorylation and AHR Nuclear Translocation Are Inhibited by Baicalein

To obtain further insight into the mechanisms by which BAI prevents the BaP/AHR-mediated CYP1A1 expression, we next examined the phosphorylation status of Src, which is known to form a protein complex with AHR in the cytoplasm. In the presence of BaP, phosphorylated Src was significantly increased compared with that of the control (2.58 ± 0.606-fold increase), which was prevented by treatment with both BAI and BaP in NHEKs (Figure 7a, Appendix A). In HaCaT cells, BAI reduced the level of phosphorylated Src compared with that in the control. The phosphorylation of Src was upregulated by BaP in HaCaT cells, which was downregulated in the presence of BAI, as in NHEKs (Figure 7b, Appendix A).

Furthermore, the status of AHR nuclear translocation was assessed, since an Src inhibitor Dasatinib reportedly prevents BaP-induced AHR nuclear translocation and inhibits the effect of BaP [62]. Dasatinib was used as a positive control, which inhibits BaP-induced AHR nuclear translocation by inhibiting Src. As shown in Figure 8a, BaP induced the nuclear translocation of AHR (Figure 8a, arrows), which was abrogated by the treatment with dasatinib or BAI. AHR nuclear translocation was confirmed with Western blotting by separately extracting cytoplasmic and nuclear protein. As shown in Figure 8b, AHR protein was significantly decreased in the cytoplasmic fraction of BaP-treated cells, whereas it was increased in the nuclear fraction of BaP-treated cells compared to that of DMSO-treated cells (1.56 ± 0.266-fold increase, Figure 8b and Appendix A). In the presence of dasatinib or BAI, BaP-induced AHR nuclear translocation was significantly inhibited. Taking all of these findings together, BAI likely prevents the deleterious effects of BaP through inhibiting Src phosphorylation and subsequent AHR nuclear translocation in keratinocytes.

## 4. Discussion

In daily life, we are exposed to environmental PAHs in tobacco smoke, diesel exhausts and industrial combustion emissions [4,63]. Among the PAHs, BaP is one of the best studied compounds, being known to induce oxidative stress, inflammation and carcinogenesis [63]. Since BaP is lipophilic, it easily crosses the cell membrane, binds to its receptor AHR, and triggers the AHR nuclear translocation, and subsequently upregulates xenobiotic metabolizing enzymes such as CYP1A1 which metabolizes BaP [6,8,9,10,17,18,19,20,21,22,23]. Resulting metabolites undergo redox cycles with generation of ROS and leading to oxidative stress [6,24,25]. The generated oxidative stress further stimulates the production of proinflammatory cytokines [7,26,27,28,29,30,31]. The flow of these reactions and emersion of hazardous effects by BaP or by other PAHs appear to be highly dependent on the AHR-CYP1A1 axis because they are attenuated in *Ahr*-deficient mice [64,65,66] and in *Cyp1a1*-deficient mice [24,25].

The inhibitory action of BAI on AHR-CYP1A1 axis has been reported by several groups [57,67]. In MCF-7 breast cancer cells treated with DMBA, one of the PAHs, enzyme kinetic analysis revealed that BAI is a competitive inhibitor of CYP1A1 activity. In addition, AHR transactivation was suppressed by BAI in the cells treated with DMBA or BaP, suggesting the ability of BAI to prevent deleterious effect of these agents through inhibition of AHR-CYP1A1 axis [57]. Furthermore, in vivo analysis using BaP-induced lung carcinogenesis mouse model, oral administration of BAI (12 mg/kg body weight/week for 16 weeks) was found to significantly inhibit the induction of BaP-induced phase I and II enzymes in the lung. Moreover, BAI treatment reduced the BaP-induced CYP1A1 expression and effectively counteracted the BaP-induced oxidative damages [67]. In accordance with these reports, our results showed that BAI is able to inhibit the BaP-induced expression of CYP1A1 in keratinocytes. In the present study, BAI was likely to competitively inhibit the binding of BaP with AHR, and to eventually downregulate AHR-CYP1A1 axis. Since AHR-CYP1A1 axis is basically a detoxifying pathway, there might be a concern that inhibition of this pathway with BAI will interrupt the detoxification of BaP and will have adverse effects. However, the toxic effects of BaP are highly dependent on the AHR-CYP1A1 axis as the toxic effects were attenuated in *Ahr*- or *Cyp1a1*-defficient mice [24,25,64,65,66]. Thus, inhibition of BaP-AHR-CYP1A1 axis by BAI may diminish “metabolism” of BaP and may down-modulate its toxic effects, because BaP per se is not unlikely to exert further toxic effects unless it is metabolized. Of note, BAI-containing crude/herbal drugs (WO and OG) also significantly inhibited BaP-induced CYP1A1 expression (Figure 2c), implying their potential medicinal use to reduce deleterious effects of BaP or other PAHs.

BaP is known to induce cytoplasmic-to-nuclear translocation of AHR in parallel with Src phosphorylation, as demonstrated previously [14,15,24]. It is also reported that the extract of a flower *Clematis apiifolia* DC suppressed the BaP-induced CYP1A1 expression and Src phosphorylation, followed by the inhibition of AHR nuclear translocation in HaCaT cells. Since dasatinib, a known Src inhibitor, also inhibited BaP-induced AHR nuclear translocation, Src is suggested to be involved in the nuclear translocation of AHR [63]. Regarding Src, BAI has been reported to dephosphorylate Src and inhibit the malignant phenotypes of non-small cell lung cancer [68]. In agreement with those reports, our results showed that BAI inhibits the BaP-induced Src phosphorylation together with the inhibition of cytoplasmic-to-nuclear translocation of AHR. These results stress an essential role of Src activation in AHR nuclear translocation and BAI is likely to inhibit the BaP-induced AHR nuclear translocation by inhibiting the Src activation.

In this context, it should be mentioned that BAI itself slightly upregulated the AHR-CYP1A1 axis in HaCaT keratinocytes, but not in NHEKs (Figure 2a,b). Although direct comparison was not appropriate, Src was likely to be more phosphorylated in immortalized HaCaT keratinocytes than in normal primary keratinocytes (Figure 7a,b). This fact may underpin the discrepancy in BAI-AHR-CYP1A1 signaling between HaCaT cells (slight responders) and NHEKs (non-responders). In parallel with this, BAI was reported to activate the AHR-CYP1A1 axis in the human HepG2 liver cancer cell line in which Src is strongly phosphorylated, even under baseline conditions [56,69].

Similar to BAI, Mohebati and colleagues reported that a polyherbal formulation, Zyflamend, and its constituent carnosol, a polyphenol, suppress BaP-induced CYP1A1 and CYP1B1 expression in HaCaT cells. Although the AHR cytoplasmic complex contains HSP90, XAP-2, and p23 in addition to Src [17], they found that Zyflamend did not change the protein amount of HSP90, XAP-2, and p23. Instead of protein expression, inhibition of HSP90 ATPase activity is proposed as a mechanism underlying the Zyflamend/carnosol-mediated AHR-CYP1A1 inhibition [70]. Since the effect of BAI on HSP90 ATPase activity has not been reported, it should be assessed in future research to reveal more detailed mechanisms by which BAI modulate AHR-CYP1A1 axis.

In addition to the effects on AHR-CYP1A1 axis, BAI is known as an effective antioxidant. In HaCaT cells, BAI protected the cells from ultraviolet B by absorbing the radiation and also by scavenging ROS [71,72]. BAI has been reported to scavenge intracellular ROS by NRF2-mediated manganese superoxide dismutase and inhibited H_2_O_2_-induced DNA damage and apoptosis [73]. BAI is also known to exert its antioxidative activity through activation of the NRF2-HMOX1 antioxidative system [73,74,75]. In parallel with this, the present study showed that BAI induced the nuclear translocation of NRF2, upregulated the expression of antioxidative enzymes and downregulated the BaP-induced ROS generation in keratinocytes. Increased oxidative stress further upregulates the production of proinflammatory cytokines by inducing the secretion of ATP and activating its receptor P2 × 7R [29,30,31]. Thus, it is speculated that BaP-induced AHR-CYP1A1-ROS signal further increased the production of IL1A and IL1B, which was successfully prevented by the induction of NRF2 antioxidative system such as HMOX1.

Considering the potent activation of NRF2-HMOX1 system, BAI or BAI-containing drugs (WO and OG) may be beneficial for the treatment of various immunoinflammatory and autoimmune diseases in which the manipulation of NRF2-HMOX1 antioxidative system is one of the potential therapeutic approaches [76,77,78,79,80]. HMOX1 degrades heme and one of the end-product CO is known to have immunomodulatory properties [76]. Using murine autoimmune hepatitis model, NRF2 pathway was shown to be involved in the development and progression of autoimmune hepatitis, and CO-releasing molecule (CORM)-A1 improved several sero-immunological and histological parameters in the model mice [76]. In rats, induction of CO by CORM-A1 improved the clinical parameters of experimental autoimmune uveoretinitis [77]. Based on a similar concept, the therapeutic potential of NRF2-HMOX1 pathway activation and CO induction has been reported in neuroinflammation [78], multiple sclerosis [79], and systemic lupus erythematosus (SLE) [80]. Indeed, Li D and colleagues explained that BAI ameliorates pristine-induced lupus nephritis via activating NRF2-HMOX1 and suppressing inflammatory cytokine production in SLE mouse model [74].

Activation of NRF2 pathway is triggered by the disruption of interaction between NRF2 and Keap1. Since Keap1/NRF2/ARE pathway is important in defensing mechanism against oxidative stress, managing this pathway may provide beneficial effects on chronic diseases such as cardiovascular diseases, diabetes, atherosclerosis, cancer, and neurodegenerative diseases [38]. In HepG2 cells, BAI was shown to target Keap1 to stimulate ubiquitination and modification of Keap1, leading to the activation of NRF2, cytoprotection and cancer chemoprevention [81]. It has also been reported that BAI protects rat pheochromocytoma cells against 6-hydroxydopamine-induced neurotoxicity through the activation of Keap1/NRF2/HMOX1 pathway, indicating the potential use of BAI to prevent neurodegenerative diseases such as Parkinson’s disease [82].

## 5. Conclusions

Taken all together, the present report reveals that BAI acts as a potent inhibitor of the AHR-CYP1A1 axis through inhibition of Src phosphorylation and following AHR nuclear translocation, and that BAI also prevents oxidative stress and proinflammatory cytokine production by scavenging ROS and activating NRF2-HMOX1 antioxidative axis in keratinocytes. BAI and BAI-containing *Scutellaria baicalensis* extracts may be beneficial as treatments for those with severe exposure to pollutants such as BaP and also for the diseases which are related to oxidative stress.

## Figures and Tables

**Figure 1 antioxidants-09-00507-f001:**
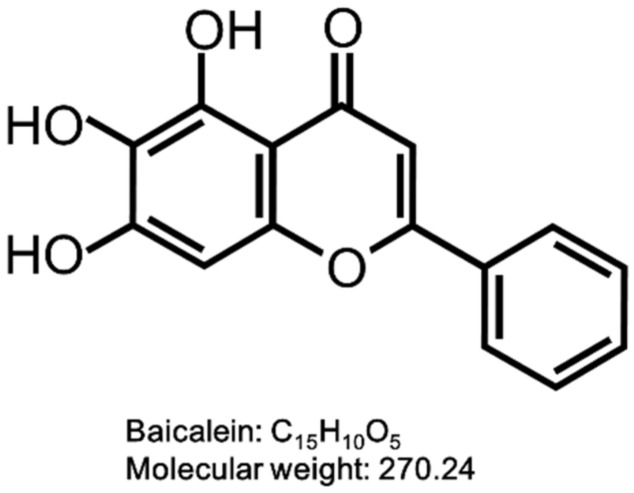
Structure of baicalein (BAI). The structure of BAI is shown.

**Figure 2 antioxidants-09-00507-f002:**
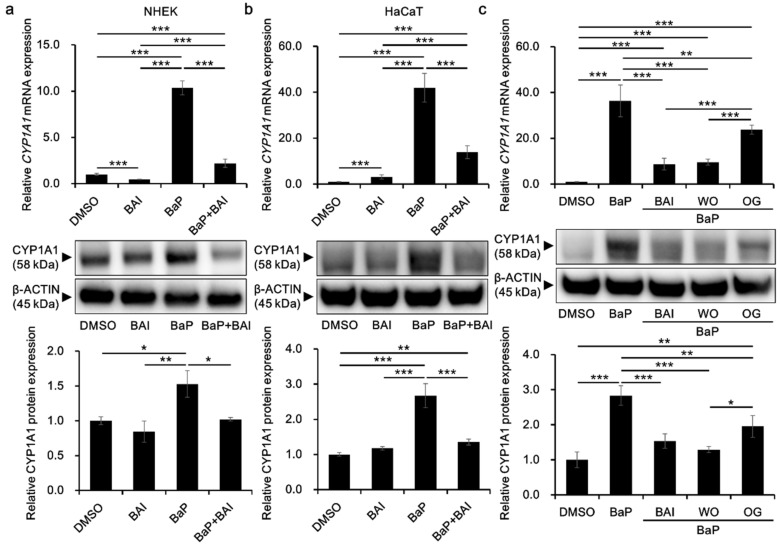
BAI inhibits BaP-induced CYP1A1 expression in keratinocytes. (**a**) NHEKs or (**b**) HaCaT cells were treated with DMSO (0.1%), BAI (10 μM for NHEKs and 25 μM for HaCaT cells), BaP (1 μM) or a combination of BAI and BaP and assessed for *CYP1A1* mRNA expression (upper panels) at 6 h- or CYP1A1 protein expression (middle and lower panels) at 24 h-post treatment. (**c**) HaCaT cells were treated with DMSO (0.1%), BAI (25 μM), BaP (1 μM) or a combination of BaP and BAI, BaP and WO, or BaP and OG, and assessed for *CYP1A1* mRNA expression (upper panels) at 6 h or CYP1A1 protein expression (middle and lower panels) at 24 h post treatment. Experiments were performed in triplicate wells and each experiment was repeated three times. Representative blot images (middle panels) and mean ± SD of CYP1A1/β-ACTIN ratio (lower panels) are shown. Whole blot images are shown in Appendix A. * *p* < 0.05, ** *p* < 0.01 and *** *p* < 0.001.

**Figure 3 antioxidants-09-00507-f003:**
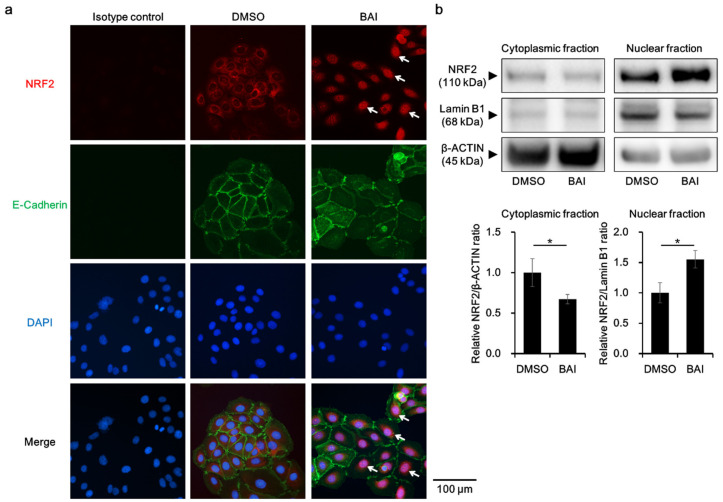
BAI induces NRF2 nuclear translocation. (**a**) HaCaT cells were treated with DMOS (0.1%) or BAI (25 μM) for 6 h and nuclear translocation of NRF2 was examined using immunocytochemistry. E-cadherin was stained simultaneously as a marker of plasma membrane. DAPI was used for nuclear staining. Experiments were repeated three times and the representative images (left panels) are shown. Scale bar = 100 μm. (**b**) HaCaT cells were treated as for (**a**) and cytoplasmic and nuclear fraction were separately extracted. β-ACTIN and Lamin B1 were used as internal controls of cytoplasmic or nuclear protein. Experiments were performed in triplicate wells and each experiment was repeated three times. Representative blot images (right upper panels) and mean ± SD of relative NRF2/β-ACTIN and NRF2/Lamin B1 ratio (right lower panels) are shown. Whole blot images are shown in Appendix A. * *p* < 0.05.

**Figure 4 antioxidants-09-00507-f004:**
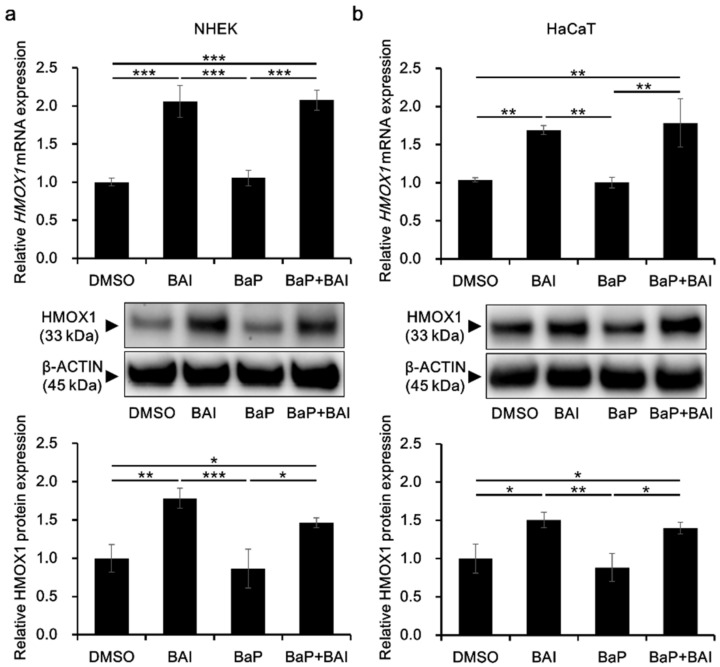
BAI inducesHMOX1 expression. (**a**) NHEKs and (**b**) HaCaT cells were treated with DMSO (0.1%), BAI (10 μM for NHEKs and 25 μM for HaCaT cells), BaP (1 μM) or a combination of BAI and BaP and assessed for *HMOX1* mRNA expression (upper panels) at 6 h or HMOX1 protein expression (middle and lower panels) at 12 h post treatment. Experiments were performed in triplicate wells and each experiment was repeated three times. Representative blot images (middle panels) and mean ± SD of relative HMOX1/β-ACTIN ratio (lower panels) are shown. Whole blot images are shown in Appendix A. * *p* < 0.05, ** *p* < 0.01 and *** *p* < 0.001.

**Figure 5 antioxidants-09-00507-f005:**
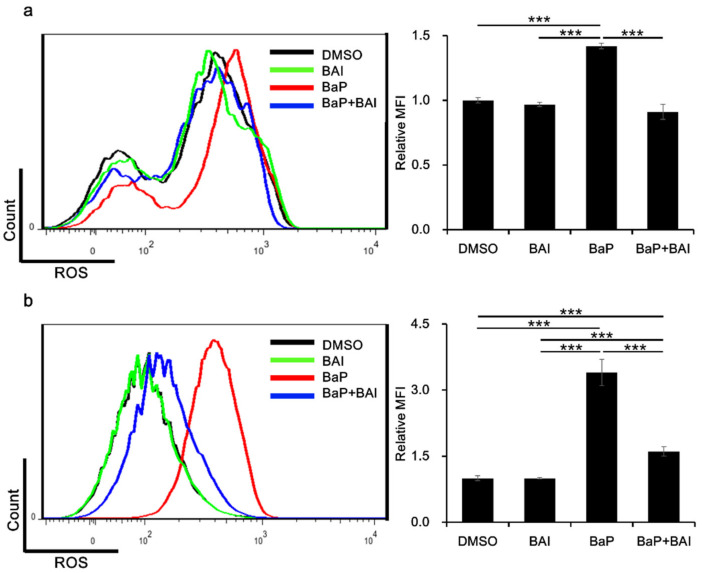
BAI exerts antioxidative effects against BaP. (**a**) NHEKs or (**b**) HaCaT cells were treated with DMSO (0.1%), BAI (10 μM for NHEKs and 25 μM for HaCaT cells), BaP (1 μM) or a combination of BAI and BaP for 12 h and ROS production was measured by flow cytometry. A representative histogram image of mean fluorescent intensity (MFI) of DCF (excitation: 480 nm, emission: 530 nm) (left) and mean ± SD of MFI (right) are shown. Experiments were performed in triplicate wells and each experiment was repeated three times. *** *p* < 0.001.

**Figure 6 antioxidants-09-00507-f006:**
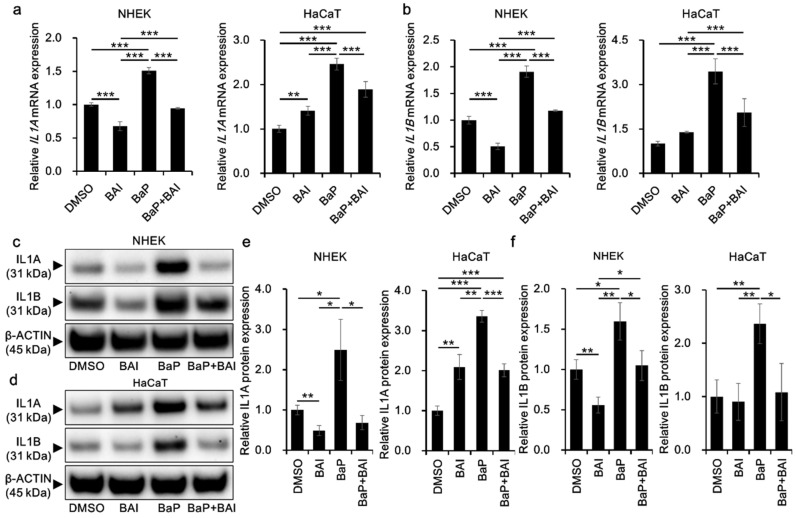
BAI inhibits BaP-induced proinflammatory cytokine expression in keratinocytes. (**a**,**b**) NHEKs and HaCaT cells were treated with DMSO (0.1%), BAI (10 μM for NHEKs and 25 μM for HaCaT cells), BaP (1 μM) or a combination of BAI and BaP for 6 h and assessed for (**a**) *IL1A* and (**b**) *IL1B* expression. (**c**–**f**) NHEKs or HaCaT cells were treated as (**a**,**b**). After 24 h, cells were harvested and protein expressions of (**c**–**e**) IL1A and (**c**,**d**,**f**) IL1B were assessed. Experiments were performed in triplicate wells and each experiment was repeated three times. Representative blot images (left lower panels) and mean ± SD of relative IL1A/β-ACTIN and IL1B/β-ACTIN ratio (right lower panels) are shown. Whole blot images are shown in Appendix A. * *p* < 0.05, ** *p* < 0.01 and *** *p* < 0.001.

**Figure 7 antioxidants-09-00507-f007:**
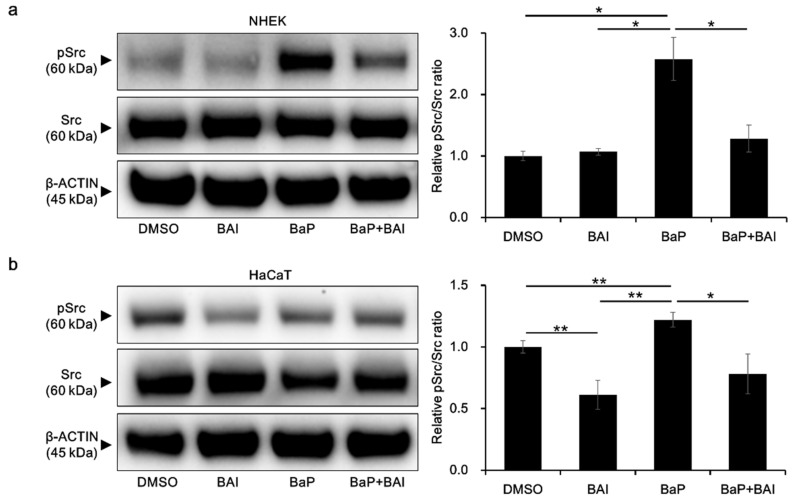
BaP-induced Src phosphorylation is inhibited by BAI. (**a**) NHEKs and (**b**) HaCaT cells were treated with DMSO (0.1%), BAI (10 μM for NHEKs and 25 μM for HaCaT cells), BaP (1 μM) or a combination of BAI and BaP for 24 h and Src phosphorylation was assessed by Western blotting. Experiments were performed in triplicate wells and each experiment was repeated three times. Representative blot images (left) and mean ± SD of relative pSrc/Src ratio (right) are shown. Whole blot images are shown in Appendix A. * *p* < 0.05 and ** *p* < 0.01.

**Figure 8 antioxidants-09-00507-f008:**
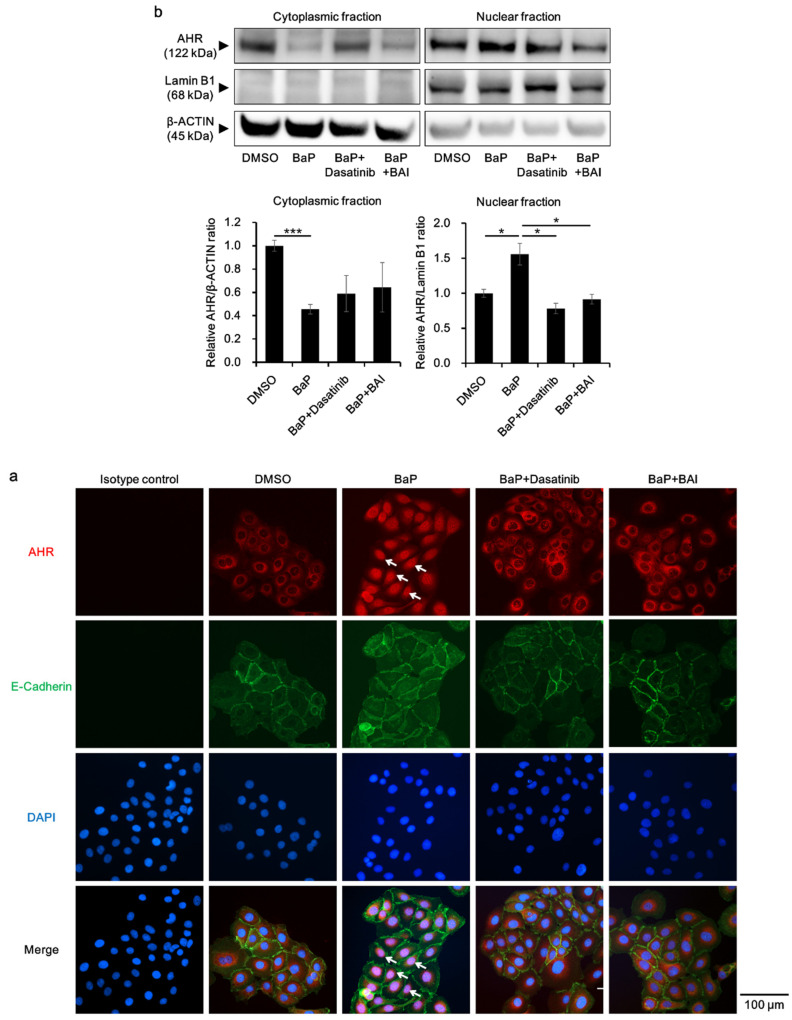
BaP-induced AHR nuclear translocation is prevented by BAI. (**a**) HaCaT cells were treated with DMSO (0.1%), BaP (1 μM), the combination of BaP and dasatinib (100 nM) or the combination of BaP and BAI (25 μM) for 24 h and nuclear translocation of AHR was examined using immunocytochemistry. E-cadherin staining indicates plasma membrane and arrows indicate AHR that translocated to the nucleus. DAPI was used for nuclear staining. Scale bar = 100 μm. (**b**) HaCaT cells were treated as in (**a**) and cytoplasmic and nuclear fraction were separately extracted. β-ACTIN and Lamin B1 were used as internal controls of cytoplasmic or nuclear protein. Representative blot images (upper panels) and mean ± SD of relative AHR/β-ACTIN and AHR/Lamin B1 ratio (lower panels) are shown. Experiments were performed in triplicate wells and each experiment was repeated three times. Whole blot images are shown in Appendix A. * *p* < 0.05 and *** *p* < 0.001.

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
