# Peer review of "Baicalein Inhibits Benzo[a]pyrene-Induced Toxic Response by Downregulating Src Phosphorylation and by Upregulating NRF2-HMOX1 System"

_antioxidants, 2020, doi:10.3390/antiox9060507_

Round 1
Reviewer 1 Report
The authors have responded convincingly and precisely to the different comments made by the reviewer.
Reviewer 2 Report
The previous version of the manuscript lacked Western blot analysis of the proteins involved. This along with other concerns raised in the previous version of the manuscript have been addressed.
This manuscript is a resubmission of an earlier submission. The following is a list of the peer review reports and author responses from that submission.
Round 1
Reviewer 1 Report
In the current study, Tanaka et al. investigated the role of Baicalein (BAI) in suppressing reactive oxygen species (ROS) generation by Benzo[a]pyrene (BaP) in primary and immortalized human keratinocyte epithelial cells. The authors concluded that BAI attenuated ROS production by stimulating the antioxidant/anti-inflammatory NRF2-HMOX1 axis and by inhibiting pro-oxidative AHR-CYP1A1 axis. While the study is interesting, below are some concerns that need to be addressed for this study to be more meaningful.
Major comments:
- One of the major limitations of this study is that the vast majority of conclusions are based on the mRNA expression data for various targets including CYP1A1, interleukins, Nrf2 and HMOX1. Since mRNA modulation does not necessarily correlate with protein expression, it is important to demonstrate if the proposed targets are similarly modulated at proteins levels.
- Immunochemistry data showing NRF2 translocation (in Fig. 5, Suppl. Fig. S4) lacks plasma membrane marker. In the absence of a plasma membrane marker, it would be difficult to ascertain subcellular compartmentalization. Therefore, a plasma membrane marker needs to be included to consolidate claims. In addition, the authors are suggested to supplement the immunostaining data by showing nuclear translocation of NRF2 and consequent HMOX1 upregulation using Western blotting.
Minor comments:
- Between Fig. 1a and 1b, there is only one treatment that is different. Authors are suggested to combine 1a and 1b into one bar graph. Likewise, Fig. 1c and 1d should also be combined as they differ by only one treatment condition.
- In Fig. 4a, combined treatment of BaP+BAI appears to reduce total Src while neither BaP nor BAI individually altered total Src level. Authors are suggested to replace this blot with a representative image to match the bar graph in Fig. 4b.
- Baicalein (BAI) was first mentioned in the abstract but was first abbreviated in the Introduction section. For convenience, baicalein should abbreviated upfront in the Abstract section. Similar change will be required for heme oxygenase 1 (HMOX1) and potentially others.
- The manuscript should be revised for the appropriate use of words. For example, the word ‘cancel’ has been used at numerous locations (lines 220, 262, 264, 267, 268, 295) in this manuscript to describe effects. Suitable alternatives are available and are encouraged to use.
Reviewer 2 Report
Reviewer comments
The CYP1A1 enzyme is induced in response to a toxic agent to proceed with its detoxification. The inhibition of this route of activation by baicalein which implications and consequences can have if benzo [a] pyrene is not detoxified? This possibility should be discussed in the manuscript.
Since Nrf2 controls the expression of antioxidant response element (ARE)-dependent genes to regulate cellular resistance to oxidants, why have the authors not evaluated the expression of some of these antioxidants?
In general, subjective expressions should be avoided: we found or we demonstrated.
Abstract
Describe the meaning of CYP1A1.
The aim and some idea about the experimental procedure should be added to the abstract.
Introduction
Additional information about the origin of benzo[a]pyrene and the mechanisms of toxicity in the body should be provided to better focus de work.
The activation of AHR in addition to the induction of CYP1A1, induces the expression of other Phase I and Phase II metabolizing enzymes in addition to modulating the response of the immune system, such as T cells. All this should be exposed and not only limited to the objective of study to Give a more complete view.
Line 45-46, the affirmation is not clear. The increase in ROS production is a consequence of the detoxification activity of CYP1A1, not a primary consequence. The sentence should be revised accordingly.
Explain the antioxidant mechanism of action of heme oxygenase 1.
Chemical characteristics of baicalein indicating that is a type of flavonoid and an image of the structure should be added.
For readers not specialized in traditional medicine, more information about Wogon and the herbal medicine Oren-gedoku-to should be provided.
Materials and methods
Briefly, explain the methodological basis of Cell Counting Kit-8.
The experimental procedure explaining the different treatments used should be clearly stated before indicating the methodology of the different parameters analysed. In addition to explaining clearly the rationale of the concentrations of benzo [a] pyrene and baicalein used.
In should be indicated the type of primary specific antibodies used indicating the company of origin, the species and the concentration in order so that the procedures can be reproducible.
Why the authors used dasatinib in the Immunocytochemistry assay? It should be explained.
How many replicates were used for each analysis?
Was the normality of the results analysed in order to apply the Student’s unpaired two-tailed t-test parametric test?
Results
Which is the meaning of “Each experiment was repeated at least three times”? The number of replicates performed should be specifically indicated.
Why in the results presented in Figure 1, are not shown in the same graph all the results by the same cell line? The results obtained for DMSO, BAI, BaP and BAI+BaP should be presented in only one graph.
It is confusing that the authors explain the results about Wogon and Oren-gedoku-to but all the information in this regard is shown as supplementary information. It would be better to introduce it in the main document, explaining in detail the treatment performed and the results obtained, or delete it and focus only with BAI.
If the results obtained for both cell types follow the same pattern, they can be explained together to avoid duplication in the text.
Why, unlike Figure 1, in Figure 2, the effects of BAI alone are not analysed? It would be interesting to see if by itself it has effects on the expression of ILA and ILB.
Why has HMOX1 expression not been analysed in treatment with BaP and in combination with BAI? It would be interesting if BaP has some kind of effect on its expression.
Why did the authors not investigate heat shock protein 90 (HSP90), hepatitis B virus X-associated protein 2 (XAP-2) and p23, and analysed c-Src? Without this information the results are incomplete.
Where the western blot data, corrected by B-actin? It should be clearly indicated.
Discussion
In the discussion, the authors simply describe and explain the results obtained by providing much general information that should be in the introduction. Further discussion should be deepened by comparing the results obtained with other studies using similar models, other PHAs or other polyphenols.
The results from the references cited in line 307, should be briefly explained and discussed.
Baicalein is capable to directly scavenge reactive oxygen species and induce antioxidant enzymes such as manganese superoxide dismutasa which can contribute to reduce the ROS production (see for example Toxicol Ind Health. 2012 Jun;28(5):412-21). All this should be discussed in the text.
Response to reviewers
Editor comments:
This study indicates that Baicalein inhibits benzo[a]pyrene-induced toxic response by downregulating Src phosphorylation and by upregulating NRF2-HMOX1 system.
The paper has some potential merits but in my opinion the message should be made much clearer by a deep description of the pleiotropic effects of the aryl hydrocarbon receptor (AHR) that are primarily, but not exclusively, antiinflammatory.
The Authors should explain in much deeper detail how the BaP-AHR-CYP1A1 axis generates reactive oxygen species (ROS) and induces IL1A 16 and IL1B upregulation and how Baicain, that is an agonist of AHR (this should be made clearer in the text as often it is indicated as AHR ligand which may be both agonist and antagonist), should counteract this pathway. By common sense it should upregulate the pathway. This needs to be explained.
Thank you very much for the suggestion. According to the editor’s comment, we have revised the sentences regarding the explanation about how the BaP-AHR-CYP1A1 axis acts and how baicalein counteracts this pathway as follows:
Line 43 to 57
In the absence of ligands, AHR localizes in the cytoplasm as a component of protein complex which consists of dimer of heat shock protein 90 (HSP90), hepatitis B virus X-associated protein 2 (XAP-2, also known as AIP or Ara9), p23 and c-Src protein kinase [10–16]. After ligand binding, XAP-2 is released from the complex and triggers conformational changes in AHR that leading to the AHR nuclear translocation by exposing its nuclear localizing signal [17]. In the nucleus, AHR dimerizes with AHR-nuclear translocator (ARNT), binds DNA response elements called xenobiotic response elements (XREs) and upregulates the transcription of phase Ⅰ and Ⅱ xenobiotic metabolizing enzymes, such as phase Ⅰ: cytochrome P450 1A1 (CYP1A1), CYP1A2, and CYP1B1 [8–10,18,19] and phase Ⅱ: NAD(P)H quinone reductase-1 (NQO1), UDP-glycosyltransferases (UGT1A1 and UGT1A6), and glutathione S-transferase (GST) [6,20–23]. When binds to AHR, BaP is oxidized by CYPs into epoxides and phenolic intermediates, those further processed by phase Ⅱ enzymes. Resulting 3,6-quinone undergoes redox cycles with generation of reactive oxygen species (ROS) and leads to oxidative stress [6,24,25]. The oxidative stress increases secretion of ATP, which in turn stimulates the production of proinflammatory cytokines such as IL-1α, IL-1β, and IL-6 [7,26–31].
Line 416 to 436
Since BaP is lipophilic, it easily crosses the cell membrane, binds to its receptor AHR, and triggers the AHR nuclear translocation, and subsequently upregulates xenobiotic metabolizing enzymes such as CYP1A1 which metabolizes BaP [6,8–10,17–23]. Resulting metabolites undergo redox cycles with generation of ROS and leading to oxidative stress [6,24,25]. The generated oxidative stress further stimulates the production of proinflammatory cytokines [7,26–31]. The flow of these reactions and emersion of hazardous effects by BaP or by other PAHs appear to be highly dependent on the AHR-CYP1A1 axis because they are attenuated in Ahr-deficient mice [64–66] and
in Cyp1a1-deficient mice [24,25].
The inhibitory action of BAI on AHR-CYP1A1 axis has been reported by several groups [57,67]. In MCF-7 breast cancer cells treated with DMBA, one of the PAHs, enzyme kinetic analysis revealed that BAI is a competitive inhibitor of CYP1A1 activity. In addition, AHR transactivation was suppressed by BAI in the cells treated with DMBA or BaP, suggesting the ability of BAI to prevent deleterious effect of these agents through inhibition of AHR-CYP1A1 axis [57,67]. Besides, in vivo analysis using BaP-induced lung carcinogenesis mouse model, oral administration of BAI (12 mg/kg body weight/week for 16 weeks) was found to significantly inhibit the induction of BaP-induced phase I and II enzymes in the lung. Moreover, BAI treatment reduced the BaP-induced CYP1A1 expression and effectively counteracted the BaP-induced oxidative damages [67]. In accordance with these reports, our results showed that BAI is able to inhibit the BaP-induced expression of CYP1A1 in keratinocytes. In the present study, BAI was likely to competitively inhibit the binding of BaP with AHR, and to eventually downregulate AHR-CYP1A1 axis.
To avoid misleading confusion it is essential that the Authors discuss the physiology of AHR much more adequately and quote adequate and updated review such as
Gutiérrez-Vázquez C1, Quintana FJ2.Regulation of the Immune Response by the Aryl Hydrocarbon Receptor.Immunity. 2018 Jan 16;48(1):19-33. doi: 10.1016/j.immuni.2017.12.012.
According to the suggestion, we have revised the sentences according to the references in introduction and discussion sections.
Line 42 to 53
AHR is a ligand-activated transcription factor and is abundantly expressed in skin cells including epidermal keratinocytes to sense environmental and endogenous chemicals [7–10]. In the absence of ligands, AHR localizes in the cytoplasm as a component of protein complex which consists of dimer of heat shock protein 90 (HSP90), hepatitis B virus X-associated protein 2 (XAP-2, also known as AIP or Ara9), p23 and c-Src protein kinase [10–16]. After ligand binding, XAP-2 is released from the complex and triggers conformational changes in AHR that leading to the AHR nuclear translocation by exposing its nuclear localizing signal [17]. In the nucleus, AHR dimerizes with AHR-nuclear translocator (ARNT), binds DNA response elements called xenobiotic response elements (XREs) and upregulates the transcription of phase Ⅰ and Ⅱ xenobiotic metabolizing enzymes, such as phase Ⅰ: cytochrome P450 1A1 (CYP1A1), CYP1A2, and CYP1B1 [8–10,18,19] and phase Ⅱ: NAD(P)H quinone reductase-1 (NQO1), UDP-glycosyltransferases (UGT1A1 and UGT1A6), and glutathione S-transferase (GST) [6,20–23].
Line 416 to 419
Since BaP is lipophilic, it easily crosses the cell membrane, binds to its receptor AHR, and triggers the AHR nuclear translocation, and subsequently upregulates xenobiotic metabolizing enzymes such as CYP1A1 which metabolizes BaP [6,8–10,17–23].
Line 465 to 470
Likewise BAI, Mohebati and colleagues reported that a polyherbal formulation Zyflamend and its constituent carnosol, a polyphenol, suppress BaP-induced CYP1A1 and CYP1B1 expression in HaCaT cells. Although the AHR cytoplasmic complex contains HSP90, XAP-2, and p23 in addition to Src [17], they found that Zyflamend did not change the protein amount of HSP90, XAP-2, and p23. Instead of protein expression, inhibition of HSP90 ATPase activity is proposed as a mechanism underlying the Zyflamend/carnosol-mediated AHR-CYP1A1 inhibition [70].
2. As the Authors prove that Baicalein upregulates the NRF2-HMOX1 system they may propose the use of Baicalein for immunoinflammatory and autoimmune diseases that benefit from upregulation of the NRF2 pathway and/or if its byproduct carbon monoxide in animal models of autoimmune hepatitis, uveitis, multiple sclerosis and SLE
Mangano K et al., Involvement of the Nrf2/HO-1/CO axis and therapeutic intervention with the CO-releasing molecule CORM-A1, in a murine model of autoimmune hepatitis.J Cell Physiol. 2018 May;233(5):4156-4165. doi: 10.1002/jcp.26223. Epub 2017 Dec 29.
Fagone P et al., Carbon monoxide-releasing molecule-A1 (CORM-A1) improves clinical signs of experimental autoimmune uveoretinitis (EAU) in rats.Clin Immunol. 2015 Apr;157(2):198-204. doi: 10.1016/j.clim.2015.02.002. Epub 2015 Feb 19.
Ângelo A. Chora et al., Heme oxygenase–1 and carbon monoxide suppress autoimmune neuroinflammation. J Clin Invest. 2007 Feb 1; 117(2): 438–447. Published online 2007 Jan 25. doi: 10.1172/JCI28844
P Fagone et al., Therapeutic potential of carbon monoxide in multiple sclerosis
Clin Exp Immunol. 2012 Feb; 167(2): 179–187. doi: 10.1111/j.1365-2249.2011.04491.x
Mackern-Oberti JP et al., Carbon monoxide exposure improves immune function in lupus-prone mice.Immunology. 2013 Sep;140(1):123-32. doi: 10.1111/imm.12124.
As the editor pointed out, we have added the proposal of use of baicalein for immunoinflammatory and autoimmune diseases in discussion section.
Line 485 to 497
Considering the potent activation of NRF2-HMOX1 system, BAI or BAI-containing drugs (WO and OG) may be beneficial for the treatment of various immunoinflammatory and autoimmune diseases in which the manipulation of NRF2-HMOX1 antioxidative system is one of the potential therapeutic approaches [76-80]. HMOX1 degrades heme and one of the end-product CO is known to have immunomodulatory properties [76]. Using murine autoimmune hepatitis model, NRF2 pathway was shown to be involved in the development and progression of autoimmune hepatitis, and CO-releasing molecule (CORM)-A1 improved several sero-immunological and histological parameters in the model mice [76]. In rats, induction of CO by CORM-A1 improved the clinical parameters of experimental autoimmune uveoretinitis [77]. Based on similar concept, therapeutic potential of NRF2-HMOX1 pathway activation and CO induction is reported in neuroinflammation [78], multiple
sclerosis [79], and systemic lupus erythematosus (SLE) [80]. Indeed, Li D and colleagues explained that BAI ameliorates pristine-induced lupus nephritis via activating NRF2-HMOX1 and suppressing inflammatory cytokine production in SLE mouse model [74].
here again, a more detailed description of the axis is required, both upstream with Keap1 and downstream and their potential utility in chronic and neurodegenerative diseases
Wenjun Tu, Hong Wang, Song Li, Qiang Liu, Hong The Anti-Inflammatory and Anti-Oxidant Mechanisms of the Keap1/Nrf2/ARE Signaling Pathway in Chronic Diseases
Aging Dis. 2019 Jun; 10(3): 637–651. Published online 2019 Jun 1. doi: 10.14336/AD.2018.0513
According to the editor’s comment, we have described the axis more in detail both upstream and downstream and their potential utility in diseases.
Line 498 to 512
Activation of NRF2 pathway is triggered by the disruption of interaction between NRF2 and Keap1. Since Keap1/NRF2/ARE pathway is important in defensing mechanism against oxidative stress, managing this pathway may provide beneficial effects on chronic diseases such as cardiovascular diseases, diabetes, atherosclerosis, cancer, and neurodegenerative diseases [38]. In HepG2 cells, BAI was shown to target Keap1 to stimulate ubiquitination and modification of Keap1, leading to the activation of NRF2, cytoprotection and cancer chemoprevention [80]. It is also reported that BAI protects rat pheochromocytoma cells against 6-hydroxydopamine-induced neurotoxicity through the activation of Keap1/NRF2/HMOX1 pathway, indicating the potential use of BAI to prevent neurodegenerative diseases such as Parkinson’s disease [82].
Taking all together, present report revealed that BAI acts as a potent inhibitor of the AHR-CYP1A1 axis through inhibition of Src phosphorylation and following AHR nuclear translocation, and that BAI also prevents oxidative stress and proinflammatory cytokine production by scavenging ROS and activating NRF2-HMOX1 antioxidative axis in keratinocytes. BAI and BAI-containing Scutellaria baicalensis extracts may be beneficial as treatments for those with severe exposure to pollutants such as BaP and also for the diseases which are related to oxidative stress.
Reviewer 1
In the current study, Tanaka et al. investigated the role of Baicalein (BAI) in suppressing reactive oxygen species (ROS) generation by Benzo[a]pyrene (BaP) in primary and immortalized human keratinocyte epithelial cells. The authors concluded that BAI attenuated ROS production by stimulating the antioxidant/anti-inflammatory NRF2-HMOX1 axis and by inhibiting pro-oxidative AHR-CYP1A1 axis. While the study is interesting, below are some concerns that need to be addressed for this study to be more meaningful.
First of all, we would like to show our appreciation to reviewers for their time to review the manuscript and for their critical and helpful comments. We have revised the manuscript based on the reviewer’s comments and explained as follows.
Major comments:
1. One of the major limitations of this study is that the vast majority of conclusions are based on the mRNA expression data for various targets including CYP1A1, interleukins, Nrf2 and HMOX1. Since mRNA modulation does not necessarily correlate with protein expression, it is important to demonstrate if the proposed targets are similarly modulated at proteins levels.
Thank you very much for the comment. We agree with the suggestion and we have evaluated the expressions of CYP1A1, interleukins, Nrf2, and HMOX1 using western blotting. We have added the results as follows: CYP1A1 (Figure 2 and Supplementary Figure S2); Nrf2 (Figure 3b and Supplementary Figure S3a); HMOX1 (Figure 4, Supplementary Figure S3b, S3c, and S4); IL1A (Figure 6c and Supplementary Figure S6); IL1B (Figure 6d and Supplementary Figure S6). The expressions of mRNA and protein showed similar pattern in this study. We also explained the results in results section.
Line 260 to 262
As well as mRNA, CYP1A1 protein expression was significantly induced by BaP and it was suppressed in the presence of BAI in both NHEKs and HaCaT cells (Figure 2a and 2b, Supplementary Figure S2).
Line 297 to 303
The expression pattern of HMOX1 protein was consistent with that of mRNA, HMOX1 was significantly induced in BAI-treated NHEKs (1.78 ± 0.131-fold increase) and HaCaT cells (1.50 ± 0.0996-fold increase) compared to DMSO-treated control (Figure 4a and 4b, middle and lower panels, Supplementary Figure S3b and S3c). BaP did not change HMOX1 expression and BAI induced HMOX1 even in the presence of BaP (Figure 4a and 4b, middle and lower panels, Supplementary Figure S3b and S3c). Similar to BAI, BAI-containing herbal drug WO and OG also induced mRNA and protein expressions of HMOX1 (Supplementary Figure S4).
Line 352 to 355
In accordance with mRNA expression, IL1A and IL1B proteins were significantly induced by BaP and BAI significantly suppressed the BaP-induced IL1A and IL1B
protein production (Figure 6c-6f, Supplementary Figure S6).
2. Immunochemistry data showing NRF2 translocation (in Fig. 5, Suppl. Fig. S4) lacks plasma membrane marker. In the absence of a plasma membrane marker, it would be difficult to ascertain subcellular compartmentalization. Therefore, a plasma membrane marker needs to be included to consolidate claims.
Based on the comments, we added E-cadherin as a plasma membrane marker and replaced the images in our revised manuscript. Figure 5 in original submission was replaced as Figure 8a and Supplementary Figure S4 in original submission was replaced as Figure 3a in the revised manuscript. As shown in Figure 3a and Figure 8a, E-cadherin staining (appears with green fluorescence) clearly indicates the plasma membrane of the cells. We believe this will help the readers to ascertain subcellular compartmentalization.
In addition, the authors are suggested to supplement the immunostaining data by showing nuclear translocation of NRF2 and consequent HMOX1 upregulation using Western blotting.
As suggested by reviewer, the nuclear translocation of NRF2 and consequent HMOX1 upregulation was confirmed by western blotting. As shown in Figure 3 and 4, induction of NRF2 nuclear translocation by baicalein (Figure 3b) and consequent HMOX1 upregulation (Figure 4) were confirmed at protein level. In addition, inhibition of BaP-induced AHR nuclear translocation by dasatinib or baicalein was also confirmed by western blotting (Figure 8b) to enhance the reliability of the result. These results are explained in results section.
Line 287 to 290
NRF2 nuclear translocation was confirmed with western blotting by separately extracting cytoplasmic and nuclear protein. As shown in Figure 3b, NRF2 protein was significantly increased in the nuclear fraction of BAI-treated cells compared to that of DMSO-treated cells (1.55 ± 0.144-fold increase, Figure 3b and Supplementary Figure S3a).
Line 394 to 400
AHR nuclear translocation was confirmed with western blotting by separately extracting cytoplasmic and nuclear protein. As shown in Figure 8b, AHR protein was significantly decreased in the cytoplasmic fraction of BaP-treated cells, whereas it was increased in the nuclear fraction of BaP-treated cells compared to that of DMSO-treated cells (1.56 ± 0.266-fold increase, Figure 8b and Supplementary Figure S8c). In the presence of dasatinib or BAI, BaP-induced AHR nuclear translocation was significantly inhibited.
Minor comments:
1. Between Fig. 1a and 1b, there is only one treatment that is different. Authors are suggested to combine 1a and 1b into one bar graph. Likewise, Fig. 1c and 1d should also be combined as they differ by only one treatment condition.
We are sorry for that the previous showing was confusing. We have combined previous Figure 1a and 1b, Figure 1c and 1d and showed as Figure 2a and Figure 2b in the revised
manuscript.
2. In Fig. 4a, combined treatment of BaP+BAI appears to reduce total Src while neither BaP nor BAI individually altered total Src level. Authors are suggested to replace this blot with a representative image to match the bar graph in Fig. 4b.
As the reviewer suggested, we have replaced the previous Figure 4a (blot images) with a representative image and showed as Figure 7a (blot images) in revised manuscript to match the result of bar graph.
3. Baicalein (BAI) was first mentioned in the abstract but was first abbreviated in the Introduction section. For convenience, baicalein should abbreviated upfront in the Abstract section. Similar change will be required for heme oxygenase 1 (HMOX1) and potentially others.
According to the instructions for author of Antioxidants which explaining as “Abbreviations should be defined in parentheses the first time they appear in the abstract, main text, and in figure or table captions and used consistently thereafter”, we have modified the abbreviations by following the instructions.
4. The manuscript should be revised for the appropriate use of words. For example, the word ‘cancel’ has been used at numerous locations (lines 220, 262, 264, 267, 268, 295) in this manuscript to describe effects. Suitable alternatives are available and are encouraged to use.
Following the suggestion, we have replaced the word ‘cancel’ with more appropriate alternatives. Lines 220, 262, 264, 267, 268, 295 in original submission were modified as lines 363, 379, 381, 391, 394, and 423, respectively in the revised manuscript.
Reviewer 2
The CYP1A1 enzyme is induced in response to a toxic agent to proceed with its detoxification. The inhibition of this route of activation by baicalein which implications and consequences can have if benzo [a] pyrene is not detoxified? This possibility should be discussed in the manuscript.
First of all, we would like to show our appreciation to reviewers for their time to review the manuscript and for their critical and helpful comments. We have revised the manuscript based on the reviewer’s comments and explained as follows.
The exhibition of toxic effects of BaP is highly dependent on the binding to AHR and subsequent CYP1A1 activation since they were attenuated in Ahr-deficient mice and in Cyp1a1-deficient mice. Thus, inhibition of BaP-AHR-CYP1A1 route by BAI may diminish “metabolism” of BaP and may down-modulate its toxic effects, because BaP per se is not unlikely to exert further toxic effects unless it is metabolized. We have added the explanation in discussion section.
Line 436 to 442
Since AHR-CYP1A1 axis is basically a detoxifying pathway, there might be a concern that inhibition of this pathway with BAI will interrupt the detoxification of BaP and will have adverse effects. However, toxic effects of BaP is highly dependent on the AHR-CYP1A1 axis as the toxic effects were attenuated in Ahr- or Cyp1a1-defficient mice [24,25,64–66]. Thus, inhibition of BaP-AHR-CYP1A1 axis by BAI may diminish “metabolism” of BaP and may down-modulate its toxic effects, because BaP per se is not unlikely to exert further toxic effects unless it is metabolized.
Since Nrf2 controls the expression of antioxidant response element (ARE)-dependent genes to regulate cellular resistance to oxidants, why have the authors not evaluated the expression of some of these antioxidants?
In original submission, we analyzed the expression of HMOX1 since it is a major downstream molecule of NRF2 and NRF2-HMOX1 pathway is crucial to protect keratinocytes from the oxidative stress. In the revised manuscript, we also evaluated the expression of other antioxidants NQO1 and GPX2 based on the reviewer’s comments and added the results in Supplementary Figure S5 and explained in Results section. The expression of NQO1 and GPX2 were upregulated by treatment with baicalein, indicating the activation of Nrf2 and its downstream antioxidants by baicalein.
Line 303 to 308
In addition to HMOX1, expressions of other antioxidantive enzymes NQO1 and GPX2 downstream of NRF2 transcription factor were assessed. As shown in Supplementary Figure S5a, BAI significantly induced NQO1 and GPX2 in NHEKs or HaCaT keratinocytes and expression patterns of NQO1 and GPX2 were similar to that of HMOX1. WO significantly induced NQO1 and GPX2 and OG significantly induced NQO1, but not GPX2 (Supplementary Figure S5b).
In general, subjective expressions should be avoided: we found or we demonstrated.
According to the reviewer’s comment, we tried to avoid the use of subjective expressions through the whole manuscript.
Abstract
Describe the meaning of CYP1A1.
The aim and some idea about the experimental procedure should be added to the abstract.
As the reviewer suggested, we have described the meaning of CYP1A1 and also added the explanation about the experimental procedure in the abstract.
Line 14 to 22
Benzo[a]pyrene (BaP), a major environmental pollutant, activates aryl hydrocarbon receptor (AHR), induces its cytoplasmic-to-nuclear translocation and upregulates the production of cytochrome P450 1A1 (CYP1A1), a xenobiotic metabolizing enzyme which metabolize BaP. The BaP-AHR-CYP1A1 axis generates reactive oxygen species (ROS) and induces proinflammatory cytokines. Although the anti-inflammatory phytochemical baicalein (BAI) is known to inhibit the BaP-AHR-mediated CYP1A1 expression, its subcellular signaling remains elusive. In this study, normal human epidermal keratinocytes and HaCaT keratinocytes were treated with BAI, BaP, or BAI + BaP, and assessed for the CYP1A1 expression, antioxidative pathways, ROS generation, and proinflammatory cytokine expressions.
Introduction
Additional information about the origin of benzo[a]pyrene and the mechanisms of toxicity in the body should be provided to better focus de work.
We agree with the comment and we have explained the origin of benzo[a]pyrene and the mechanisms of its toxicity through AHR in introduction section.
Line 37 to 41
BaP is emitted to the environment mostly from inefficient combustion of coal or from other industrial plants, and also from exhaust gas of vehicles such as automobile and airplane [4]. Once drawn into the body, BaP binds to a chemical sensor aryl hydrocarbon receptor (AHR) and is metabolized, and the metabolites trigger the following reactions that eventually damage the cells [5,6].
Line 53 to 57
When binds to AHR, BaP is oxidized by CYPs into epoxides and phenolic intermediates, those further processed by phase Ⅱ enzymes. Resulting 3,6-quinone undergoes redox cycles with generation of reactive oxygen species (ROS) and leads to oxidative stress [6,24,25]. The oxidative stress increases secretion of ATP, which in turn stimulates the production of proinflammatory cytokines such as IL-1α, IL-1β, and IL-6 [7,26–31].
The activation of AHR in addition to the induction of CYP1A1, induces the expression of other Phase I and Phase II metabolizing enzymes in addition to modulating the response of the immune system, such as T cells. All this should be exposed and not only limited to the objective of study to give a more complete view.
As the reviewer pointed out, we have added the explanation regarding Phase I and Phase II metabolizing enzymes which are induced after the AHR activation in introduction section.
Line 48 to 53
In the nucleus, AHR dimerizes with AHR-nuclear translocator (ARNT), binds DNA
response elements called xenobiotic response elements (XREs) and upregulates the transcription of phase Ⅰ and Ⅱ xenobiotic metabolizing enzymes, such as phase Ⅰ: cytochrome P450 1A1 (CYP1A1), CYP1A2, and CYP1B1 [8–10,18,19] and phase Ⅱ: NAD(P)H quinone reductase-1 (NQO1), UDP-glycosyltransferases (UGT1A1 and UGT1A6), and glutathione S-transferase (GST) [6,20–23].
Line 45-46, the affirmation is not clear. The increase in ROS production is a consequence of the detoxification activity of CYP1A1, not a primary consequence. The sentence should be revised accordingly.
We apologize that original explanation was not clear. As the reviewer pointed out, we have revised the sentences accordingly in introduction section.
Line 53 to 57
When binds to AHR, BaP is oxidized by CYPs into epoxides and phenolic intermediates, those further processed by phase Ⅱ enzymes. Resulting 3,6-quinone undergoes redox cycles with generation of reactive oxygen species (ROS) and leads to oxidative stress [6,24,25]. The oxidative stress increases secretion of ATP, which in turn stimulates the production of proinflammatory cytokines such as IL-1α, IL-1β, and IL-6 [7,26–31].
Explain the antioxidant mechanism of action of heme oxygenase 1.
According to the comment, explanation of the antioxidant mechanism of heme oxygenase 1 was added in introduction section.
Line 68 to 69
HMOX1 catalyzes heme to release free iron to form biliverdin which further metabolizes to carbon monoxide (CO) and bilirubin, and exerts antioxidative effect [38].
Chemical characteristics of baicalein indicating that is a type of flavonoid and an image of the structure should be added.
As reviewer pointed out, baicalein is a type of flavonoid found in the root of Scutellaria baicalensis. We have added the image of the structure as Figure 1 and explained in the Introduction section.
Line 73 to 74
It is also known as a plant-derived AHR agonist with the structure of flavonoid and also the aglycone of the flavone glycoside baicalin (Figure 1).
For readers not specialized in traditional medicine, more information about Wogon and the herbal medicine Oren-gedoku-to should be provided.
We agree with the reviewer and provided more information and added references about Wogon and O-ren-gedoku-to to make it more kind to the readers.
Line 76 to 82
BAI is contained in the traditional crude drug Wogon (WO), which is made from the dried root of Scutellaria baicalensis. WO contains active compounds bailcalin and wogonin in addition to BAI, and is further mixed with other crude drugs to compose
several herbal medicines, such as Oren-gedoku-to (OG, Huanglian-Jie-Du-Tang in Chinese). OG is used for alleviating the symptoms of various diseases including cerebrovascular disease, gastritis, liver dysfunction, dermatitis, inflammation, and so on [52–55].
Materials and methods
Briefly, explain the methodological basis of Cell Counting Kit-8.
According to the reviewer’s comment, explanation of methodological basis of Cell Counting Kit-8 was added to the materials and methods section, subsection 2.3. Cell proliferation assay.
Line 117 to 120
After incubation with chemicals, cells were treated with CCK-8 solution for 2-4 h which contains WST-8 coloring reagent. In live cells, WST-8 is reduced to WST-8 formazan (orange color with maximum absorption around 450 nm) and the amount of WST-8 formazan is proportional to the live cell number.
The experimental procedure explaining the different treatments used should be clearly stated before indicating the methodology of the different parameters analysed.
According to the suggestion, we have modified the overall materials and methods section, explaining the experimental procedure of different treatments used.
In addition to explaining clearly the rationale of the concentrations of benzo [a] pyrene and baicalein used.
The concentration of benzo[a]pyrene was determined based on the previous report from our research group using the same cell type and methods of analyses, showing that the benzo[a]pyrene at 1 μM is sufficient to activate AHR. We have added reference and explained in results section. Regarding baicalein, we have explained the rationale of its concentration in the original submission. We first tested the toxic concentration of baicalein in NHEKs or HaCaT cells (Supplementary Fig. 1) and selected the highest concentration (10 μM for NHEKs and 25 μM for HaCaT cells) which does not show toxic effects in our assay. The explanation is indicated in results section.
Line 250 to 254
The concentration of BaP was determined based on the previous report using the same cell type and methods of analyses, and the concentration is sufficient to activate AHR [58].
BAI did not affect cell proliferation up to 10 μM in NHEKs, and up to 25 μM in HaCaT cells, an immortalized human keratinocyte cell line [59] (Supplementary Figure S1). Thus, BAI at 10 μM for NHEKs and 25 μM for HaCaT cells were used in the following experiments.
In should be indicated the type of primary specific antibodies used indicating the company of origin, the species and the concentration in order so that the procedures can be reproducible.
The information of primary antibodies were originally summarized in Materials and
Methods section, subsection 2.1. Reagents. In the revised manuscript, we have moved these information to subsection 2.8. Western blotting and 2.10. Immunocytochemistry, so that the readers can easily find the company of origin, the species, and the concentration of antibodies used.
Line 188 to 198
2.8. Western blotting
The primary antibodies used were as follows: rabbit anti-human Src antibody (#2123, 1:2,000), rabbit anti-human phosphor-Src (Tyr416) antibody (#2101, 1:1,000), rabbit anti-human β-actin antibody (#4970, 1:1,000) (all purchased from Cell Signaling Technology, Danvers, MA, USA), rabbit anti-HO-1/HMOX1 (10701-AP, 1:1,000, Proteintech, Rosemont, IL, USA), rabbit anti-CYP1A1 antibody (ab3568, 1:500, Abcam), rabbit anti-IL-1 alpha antibody (ab9614, 1:1,000, Abcam), rabbit IL-1β antibody (#12703, 1:1,000, Cell Signaling Technology), rabbit anti-human AHR (sc-5579, 1:1,000, Santa Cruz Biotechnology, Inc., Dallas, TX, USA), and rabbit anti-human NRF2 (sc-13032, 1:1,000, Santa Cruz Biotechnology, Inc.). After three washes with 0.1% TBS-T, membranes were further treated with horseradish peroxidase (HRP)-conjugated secondary antibody (HRP-linked goat anti-rabbit IgG (#7074, 1:10,000, Cell Signaling Technologies)) diluted in Can Get Signal® Immunoreaction Enhancer Solution 2 (Toyobo Co., Ltd.) at room temperature for 60 min.
Line 229 to 234
2.10. Immunocytochemistry
The primary antibodies used were as follows: rabbit anti-human AHR (sc-5579, 1:100, Santa Cruz Biotechnology, Inc.), rabbit anti-human NRF2 (sc-13032, 1:100, Santa Cruz Biotechnology, Inc.), and mouse anti-human E-cadherin (610181, 1:200, BD Biosciences, San Jose, CA, USA). Cells were then washed three times with DPBS and further treated with Alexa Fluor® 546-conjugated goat anti-rabbit IgG (H + L) (A11010, 1:400, Thermo Fisher Scientific), and Alexa Fluor® 488-conjugated goat anti-mouse IgG (H + L) (A11001, 1:400, Thermo Fisher Scientific) secondary antibodies in dark.
Why the authors used dasatinib in the Immunocytochemistry assay? It should be explained.
Datatinib is a well-known Src inhibitor and it reportedly inhibits BaP-induced nuclear translocation of AHR. Then in our study we used dasatinib as a positive control which is a Src inhibitor and it inhibits BaP-induced AHR nuclear translocation. We have explained it in results section.
Line 390 to 393
Furthermore, the status of AHR nuclear translocation was assessed since a Src inhibitor Dasatinib reportedly prevents BaP-induced AHR nuclear translocation and inhibits the effect of BaP [62]. Dasatinib was used as a positive control which inhibits BaP-induced AHR nuclear translocation by inhibiting Src.
How many replicates were used for each analysis?
We are sorry that our explanation in original submission was incomplete. Each experiment were performed in triplicate wells and repeated 3 times. We have added the explanation in materials and methods section, subsection 2.11. Statistical analysis.
Line 240
Each experiment was performed in triplicate wells and repeated three times.
Was the normality of the results analysed in order to apply the Student’s unpaired two-tailed t-test parametric test?
We used Shapiro-Wilk test to analyze the normality of the values. We have added the explanation in materials and methods section, subsection 2.11. Statistical analysis.
Line 244 to 245
Shapiro-Wilk test was used to test normality of the samples.
Results
Which is the meaning of “Each experiment was repeated at least three times”? The number of replicates performed should be specifically indicated.
As we explained above, each experiment were performed in triplicate wells and repeated 3 times. We have added the explanation in Materials and Methods section, subsection 2.11. Statistical analysis.
Line 240
Each experiment was performed in triplicate wells and repeated three times.
Why in the results presented in Figure 1, are not shown in the same graph all the results by the same cell line? The results obtained for DMSO, BAI, BaP and BAI+BaP should be presented in only one graph.
According to the comment, we have combined the results of Fig. 1a and 1b, Fig. 1c and 1d to avoid confusion and showing as Figure 2a and Figure 2b in revised manuscript.
It is confusing that the authors explain the results about Wogon and Oren-gedoku-to but all the information in this regard is shown as supplementary information. It would be better to introduce it in the main document, explaining in detail the treatment performed and the results obtained, or delete it and focus only with BAI.
Thank you for the suggestion. We moved the results using Wogon and Oren-gedoku-to to the main results as Figure 2c and explained in results section.
Line 264 to 271
In addition, since BAI is a constituent of the crude drug WO and the WO-containing herbal drug OG, it was further tested whether these phytodrugs could inhibit BaP-induced CYP1A1 expression as did BAI. WO and OG were prepared in the boiled water (at 50°C for 60 min) and added to the culture medium with BaP. As shown in Figure 2c, both WO and OG significantly reduced the BaP-induced CYP1A1 expression at mRNA and protein levels (Figure 2c and Supplementary Figure S2). WO caused stronger suppression of CYP1A1 than that of OG and its effect was comparable to that of BAI at mRNA and protein levels. These results imply that BAI and BAI-containing drugs may protect keratinocytes from the hazardous effects of BaP.
If the results obtained for both cell types follow the same pattern, they can be explained together to avoid duplication in the text.
Following the reviewer’s suggestion, we revised sentences throughout the results section, to explain the results together when both cell types follow the same pattern.
Why, unlike Figure 1, in Figure 2, the effects of BAI alone are not analysed? It would be interesting to see if by itself it has effects on the expression of ILA and ILB.
We have analyzed the single effect of BAI on IL1A and IL1B and showed as Figure 6 (Supplementary Figure S6 for whole western blotting image) in revised manuscript and explained in results section. Single treatment of BAI significantly downregulated the mRNA/protein expressions of IL1A and IL1B in NHEKs. On the other hand, IL1A expression was slightly but significantly upregulated by BAI in HaCaT cells, although the level was lower than that of BaP. IL1B was not altered by BAI in HaCaT cells.
Line 355 to 358
Single treatment of BAI significantly downregulated the mRNA/protein expressions of IL1A and IL1B in NHKEs (Figure 6). On the other hand, IL1A expression was slightly but significantly upregulated by BAI in HaCaT cells, although the level was lower than that of BaP. IL1B was not altered by BAI in HaCaT cells (Figure 6).
Why has HMOX1 expression not been analysed in treatment with BaP and in combination with BAI? It would be interesting if BaP has some kind of effect on its expression.
The effect of BaP on HMOX1 was analyzed and it did not affect the expression of HMOX1 in our experimental condition. In combination with BAI, the expression of HMOX1 was significantly increased compared to that of DMSO-treated control. We have added the results as Figure 4 (Supplementary Figure S3b and S3c for whole western blotting image) and explained in results section.
Line 294 to 302
As shown in Figure 4, treatment with BAI significantly induced the HMOX1 mRNA expression in NHEKs (2.06 ± 0.210-fold increase) and in HaCaT cells (1.69 ± 0.0590-fold increase). BaP did not affect the HMOX1 expression and BAI was able to induce HMOX1 even in the presence of BaP. The expression pattern of HMOX1 protein was consistent with that of mRNA, HMOX1 was significantly induced in BAI-treated NHEKs (1.78 ± 0.131-fold increase) and HaCaT cells (1.50 ± 0.0996-fold increase) compared to DMSO-treated control (Figure 4a and 4b, middle and lower panels, Supplementary Figure S3b and S3c). BaP did not change HMOX1 expression and BAI induced HMOX1 even in the presence of BaP (Figure 4a and 4b, middle and lower panels, Supplementary Figure S3b and S3c).
Why did the authors not investigate heat shock protein 90 (HSP90), hepatitis B virus X-associated protein 2 (XAP-2) and p23, and analysed c-Src? Without this information the results are incomplete.
Thank you very much for the comment. It is reported that polyherval formulation Zyflamend and its constituent carnosol, a polyphenol, suppress BaP-induced CYPs in HaCaT cells, unaccompanied by the change of protein amount of HSP90, XAP-2, and p23. Instead, the inhibition of HSP90 ATPase activity is proposed as a mechanism underlying the AHR-CYP1A1 inhibition. Thus, we did not investigate the expressions of
HSP90, XAP-2, and p23 in current study and since it has not been reported about the effect of BAI on HSP90 ATPase activity, it should be assessed in the future research to reveal more detailed mechanisms by which BAI modulate AHR-CYP1A1 axis. We have added the explanation is discussion section.
Line 465 to 472
Likewise BAI, Mohebati and colleagues reported that a polyherbal formulation Zyflamend and its constituent carnosol, a polyphenol, suppress BaP-induced CYP1A1 and CYP1B1 expression in HaCaT cells. Although the AHR cytoplasmic complex contains HSP90, XAP-2, and p23 in addition to Src [17], they found that Zyflamend did not change the protein amount of HSP90, XAP-2, and p23. Instead of protein expression, inhibition of HSP90 ATPase activity is proposed as a mechanism underlying the Zyflamend/carnosol-mediated AHR-CYP1A1 inhibition [70]. Since it has not been reported about the effect of BAI on HSP90 ATPase activity, it should be assessed in the future research to reveal more detailed mechanisms by which BAI modulate AHR-CYP1A1 axis.
Where the western blot data, corrected by B-actin? It should be clearly indicated.
We are so sorry about our mistake that the results were not corrected by β-actin in original submission since pSrc/Src ratio were directly calculated from the values of pSrc and Src. In the revised manuscript, β-actin was utilized to normalize the values of CYP1A1, NRF2, HMOX1, IL1A, IL1B, and AHR in western blotting corresponding to Figure 2 (CYP1A1), Figure 3b (NRF2), Figure 4 (HMOX1), Supplementary Figure S4b (HMOX1), Figure 6c (IL1A) and 6d (IL1B), Figure 8b (AHR), and Supplementary Figure S7b (AHR). We have indicated about the data correction by β-actin in each figure legends and added the explanation in materials and methods section too.
Line 202 to 204
The values of whole and cytoplasmic protein (CYP1A1, NRF2, HMOX1, IL1A, IL1B, and AHR) were normalized with the value of β-ACTIN and the values of nuclear protein (Lamin B1, NRF2, and AHR) were normalized with the value of Lamin B1.
Discussion
In the discussion, the authors simply describe and explain the results obtained by providing much general information that should be in the introduction. Further discussion should be deepened by comparing the results obtained with other studies using similar models, other PHAs or other polyphenols.
As reviewer suggested we have revised whole discussion section by adding discussion comparing our results with other studies.
The results from the references cited in line 307, should be briefly explained and discussed.
We have added the explanation about the reference (reference 41 in original submission and reference 57 in revised manuscript) in Discussion section. References 35 and 42 in original submissions were removed from there since they are not directly explaining the inhibition of BaP-induced AHR-CYP1A1 expression by BAI.
Line 425 to 429
The inhibitory action of BAI on AHR-CYP1A1 axis has been reported by several groups [57,67]. In MCF-7 breast cancer cells treated with DMBA, one of the PAHs, enzyme kinetic analysis revealed that BAI is a competitive inhibitor of CYP1A1 activity. In addition, AHR transactivation was suppressed by BAI in the cells treated with DMBA or BaP, suggesting the ability of BAI to prevent deleterious effect of these agents through inhibition of AHR-CYP1A1 axis [57].
Baicalein is capable to directly scavenge reactive oxygen species and induce antioxidant enzymes such as manganese superoxide dismutasa which can contribute to reduce the ROS production (see for example Toxicol Ind Health. 2012 Jun;28(5):412-21). All this should be discussed in the text.
According to the comment, we have added the sentences regarding ROS scavenging effect of BAI in discussion section.
Line 473 to 478
In addition to the effects on AHR-CYP1A1 axis, BAI is known as an effective antioxidant. In HaCaT cells, BAI protected the cells from ultra violet B by absorbing the radiation and also by scavenging ROS [71,72]. BAI is reported to scavenge intracellular ROS by NRF2-mediated manganese superoxide dismutase and inhibited H2O2-induced DNA damage and apoptosis [73]. BAI is also known to exert its antioxidative activity through activation of the NRF2-HMOX1 antioxidative system [73–75].